# AdaSearch: Balancing Parametric Knowledge and Search in Large Language Models via Reinforcement Learning

## Abstract

Equipping large language models (LLMs) with search engines via reinforcement learning (RL) has emerged as an effective approach for building search agents. However, overreliance on search introduces unnecessary cost and risks exposure to noisy or malicious content, while relying solely on parametric knowledge risks hallucination. The central challenge is to develop agents that adaptively balance parametric knowledge with external search, invoking search only when necessary. Prior work mitigates search overuse by shaping rewards around the number of tool calls. However, these penalties require substantial reward engineering, provide ambiguous credit assignment, and can be exploited by agents that superficially reduce calls. Moreover, evaluating performance solely through call counts conflates necessary and unnecessary search, obscuring the measurement of true adaptive behavior. To address these limitations, we first quantify the self-knowledge awareness of existing search agents via an F1-based decision metric, revealing that methods such as Search-R1 often overlook readily available parametric knowledge. Motivated by these findings, we propose AdaSearch, a simple two-stage, outcome-driven RL framework that disentangles problem solving from the decision of whether to invoke search, and makes this decision process explicit and interpretable. This transparency is crucial for high-stakes domains such as finance and medical question answering, yet is largely neglected by prior approaches. Experiments across multiple model families and sizes demonstrate that AdaSearch substantially improves knowledge-boundary awareness, reduces unnecessary search calls, preserves strong task performance, and offers more transparent, interpretable decision behaviors.

## 1 Introduction

Large language models (LLMs) have achieved significant advances across various natural language processing tasks (Brown et al., 2020; Hendrycks et al., 2020; Team et al., 2023; Touvron et al., 2023; Wei et al., 2022; Guo et al., 2025). Yet they face inherent limitations: parametric knowledge alone cannot fully capture domain-specific (Lewis et al., 2020; Yang et al., 2024) or rapidly evolving information (Kasai et al., 2023; Vu et al., 2024), and models may hallucinate when queried beyond their knowledge boundary (Ji et al., 2023; Xu et al., 2024; Huang et al., 2025a; Kalai et al., 2025). To overcome these challenges, integrating LLMs with external knowledge sources has become a crucial direction (Karpukhin et al., 2020; Guu et al., 2020; Nakano et al., 2021; Lazaridou et al., 2022; Shi et al., 2024; Press et al., 2023; Nakano et al., 2021; Feng et al., 2024).

Early work has primarily focused on retrieval-augmented generation (RAG) (Lewis et al., 2020; Borgeaud et al., 2022), which searches for relevant passages and appends them to the context before generation (Shuster et al., 2021; Ram et al., 2023; Jiang et al., 2023; Asai et al., 2024; Wei et al., 2024). However, the relatively static pipeline of RAG overlooks the model's reasoning process, yielding suboptimal results (Trivedi et al., 2023). Moreover, enabling dynamic multi-turn search is challenging due to the requirement of large-scale annotated trajectories and the non-differentiable nature of the search operation (Schick et al., 2023; Asai et al., 2024; Guan et al., 2025). To address these limitations, treating search as a tool and training tool-integrated LLMs via reinforcement learning (RL) has emerged as a promising paradigm in recent works (Chen et al., 2025b; Jin et al., 2025a;

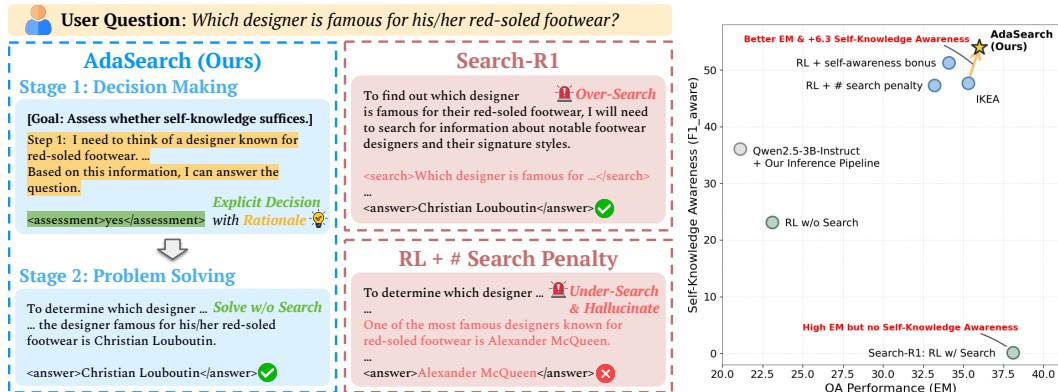

Figure 1: Comparison of RL methods for search agents. *Left*: ADASEARCH provides more transparent and interpretable decisions about whether to search. *Right*: ADASEARCH achieves the best overall self-knowledge awareness while preserving task performance.

Wang et al., 2025; Huang et al., 2025b; Fan et al., 2025). In such setting, models are trained to act as search agents, interleaving reasoning with search in a multi-turn, interactive fashion. This allows LLMs to issue queries adaptively and integrate retrieved information into their reasoning process, outperforming prior prompting-based and supervised approaches.

Nevertheless, building search agents that are truly adaptive remains challenging (Jeong et al., 2024; Qian et al., 2025; Huang et al., 2025b; Wang et al., 2025). Ideally, a search agent should balance parametric knowledge and invoke external search only when necessary. Excessive search calls might raise concerns about efficiency, safety, and privacy, potentially increasing unnecessary costs, exposing the agent to noisy or malicious content, and unintentionally leaking sensitive information. Prior attempts (Huang et al., 2025b; Wang et al., 2025) encourage adaptivity and mitigate overuse by tying rewards and evaluation to the number of search invocations. However, reward shaping around search-call counts requires careful engineering (Table 3), often involving substantial human effort and trial-and-error to determine suitable penalties for redundant search. Furthermore, reward credit assignment (Pignatelli et al., 2023) may be ambiguous: agents may exploit these signals by crafting stronger queries to reduce calls or by superficially avoiding calls even when search is necessary. In addition, naively employing search-call counts as an evaluation metric conflates necessary and unnecessary calls, obscuring the measurement of true adaptive behavior.

Beyond efficiency, a key limitation of prior search-agent frameworks is the lack of *transparency and interpretability* in the decision to invoke search. Because existing methods provide no explicit supervision for this decision, the reasoning process behind "*Should I search?*" remains implicit and difficult for users to interpret. This limitation becomes critical in real-world, high-stakes environments such as finance, medical question answering, and legal compliance, where users must understand why an agent chooses to rely on parametric knowledge or to consult external information. Consider a case where the agent skips search but then produces an incorrect answer: without an explicit decision rationale, users cannot determine whether the model was overconfident in its knowledge, whether it misinterpreted the query, or whether the failure arose from other factors. Such opacity hinders trust, auditing, and safe deployment of search agents.

In this work, we introduce ADASEARCH, a simple outcome-based RL framework that disentangles and jointly optimizes two abilities: *problem solving* and *deciding whether to invoke search*. Specifically, during training we adopt different prompts to guide LLMs in three modes: *(i)* solving the problem with internal parametric knowledge, *(ii)* solving with the problem external search, and *(iii)* explicitly deciding whether search invocation is needed. At inference time, the model first decides whether the question can be solved using its own knowledge; if so, the model answers the question directly, otherwise search calls will be invoked as part of the model's reasoning process (Figure 2).

Unlike prior approaches that depend on intricate reward engineering and often suffer from ambiguous credit assignment and implicit decision behavior, ADASEARCH simply leverages task outcomes as rewards with a training framework that provides clearer learning signals, avoids the pitfalls of ambiguous tool-count penalties, and enhances the model's self-knowledge awareness through ex-

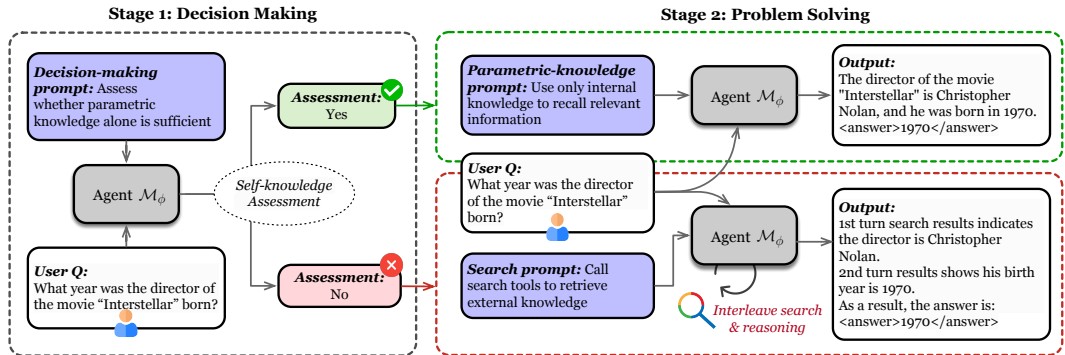

Figure 2: Overview of our proposed ADASEARCH framework. In stage 1, the agent explicitly reasons to decide whether the query can be solved using parametric knowledge. In stage 2, it follows the parametric-knowledge prompt if the knowledge is sufficient; otherwise, it switches to the search prompt to interleave reasoning and search for the final answer.

plicit reasoning (Figure 1 left). Experiments and evaluations with our proposed fine-grained self-knowledge awareness F1 metric demonstrate that ADASEARCH fosters better adaptivity, reducing unnecessary search calls while preserving strong task performance (Figure 1 right).

In summary, our contributions are three-fold:

- We propose **ADASEARCH**, a simple yet effective multi-task RL framework that explicitly optimizes both problem solving and decision making to build adaptive search agents.

- ADASEARCH improves interpretability by generating explicit reasoning during the decision stage, enhancing the trustworthiness of search agents in real-world scenarios.

- ADASEARCH eliminates the need for complex reward engineering while outperforming existing methods in both problem-solving accuracy and self-knowledge awareness, and generalizes across different model sizes and families.

## 2 EXAMINE SELF-KNOWLEDGE AWARENESS ON VANILLA SEARCH AGENTS

### 2.1 SELF-KNOWLEDGE AWARENESS F1 SCORE

We evaluate models' awareness on parametric knowledge by computing the F1 score, where the positive class corresponds to the model deciding that its parametric knowledge is sufficient to answer the input question (equivalently, not invoking search). For brevity, we use the notation $\mathbf{F1}_{\text{aware}}$ in the following sections. To determine whether a test instance is a true or false sample, we check whether the model can in fact solve the instance using parametric knowledge alone.

Formally, we define

$$\mathbf{F1}_{\text{aware}} = \frac{2 \cdot \text{Precision} \cdot \text{Recall}}{\text{Precision} + \text{Recall}}, \tag{1}$$

where precision and recall are computed on the binary decision of whether to use search, comparing the model's choice with an oracle label that indicates whether the question is solvable with parametric knowledge. This metric reflects model's awareness on its parametric knowledge.

Previous RL-based methods (Jin et al., 2025a; Wang et al., 2025; Huang et al., 2025b) adopt a single prompt that asks the agent to conduct the decision-making process and problem-solving simultaneously. To compute the score for these methods, two cases arise: (1) if the model does not call search, we directly evaluate the EM of its final answer to determine whether the instance is a true or false sample; (2) if the model does call search, we enforce parametric-only reasoning by applying another system prompt $s_{\text{param}}$ (Appendix E), and use the correctness of the resulting answer to decide whether the instance is a true or false sample.

## 2.2 Underuse of Parametric Knowledge in Vanilla Search Agents

In our preliminary experiments, we evaluate prior outcome-based RL methods using Qwen2.5-7B-Base (Team, 2024). Specifically, we include (1) Search-R1 (Jin et al., 2025a), which equips LLMs with search access and optimizes them with outcome rewards, and (2) RL w/o Search, which conducts RL training without search access and fully relies on parametric knowledge. QA performance is measured by exact match (**EM**). We follow the testing setup of Jin et al. (2025a) and report macro-averaged **EM**, **F1**$_\text{aware}$, and confusion matrices. We re-evaluate the official Search-R1 checkpoint[1]. For RL w/o Search, we report the R1-base results from Jin et al. (2025a).

The results are summarized in Table 1. For Search-R1, although search access improves **EM**, its **F1**$_\text{aware}$ is 0 because the model invokes search for every query. This overreliance, reflected in many false negatives, exemplifies tool overuse (Qian et al., 2025), where the model searches even when parametric knowledge suffices. In contrast, the model trained without search access achieves a higher **F1**$_\text{aware}$ but lower **EM** due to limited internal knowledge. These observations highlight the central challenge: enabling greater adaptivity by balancing parametric and external knowledge.

Table 1: Macro-averaged QA performance (EM), self-knowledge awareness (**F1**$_\text{aware}$), and confusion matrix results on Qwen2.5-7B-Base. **TP**: true positives; **TN**: true negatives; **FP**: false positives; **FN**: false negatives.

| Method | EM | F1$_\text{aware}$ | TP | TN | FP | FN |
|---|---|---|---|---|---|---|
| Search-R1 | 39.4 | 0.0 | 0.0 | 72.4 | 0.0 | 27.6 |
| RL w/o Search | 27.6 | 27.6 | 27.6 | 0.0 | 72.4 | 0.0 |

## 3 Our Method: AdaSearch

We introduce ADASEARCH, illustrated in Figure 2. The key idea is to achieve adaptivity without complex reward engineering by disentangling problem solving from the decision of whether to invoke search. Separating these sub-tasks and optimizing them independently enables training policy LLM with simple outcome-based rewards. In this section, we first present the problem formulation (Sec. 3.1), and then describe the ADASEARCH framework (Sec. 3.2). Our main approach is a two-stage training framework, where the first stage targets problem solving and the second stage focuses on decision making. Finally, we conclude with the inference pipeline of ADASEARCH (Sec. 3.3).

### 3.1 Problem Formulation

The problem setting is the same as Jin et al. (2025a). Given an input $x$, we let policy LLM $\pi_\theta$ interact with the search engine $\mathcal{E}$ to generate a problem solving trajectory $\tau$. Formally,

$$\tau \sim \pi_\theta(\cdot \mid x; \mathcal{E}). \tag{2}$$

Our goal is to train the policy $\pi_\theta$ to interact with $\mathcal{E}$ by maximizing the following RL objective:

$$J(\theta) = \max_{\pi_\theta} \mathbb{E}_{(x,y)\sim\mathcal{D}, \tau\sim\pi_\theta(\cdot|x;\mathcal{E})} \left[R(\tau, y)\right] - \beta \, \mathbb{D}_\text{KL}\left(\pi_\theta(\cdot \mid x; \mathcal{E}) \| \pi_\text{ref}(\cdot \mid x; \mathcal{E})\right), \tag{3}$$

where $\mathcal{D} = \{(x_i, y_i)\}_{i=1}^D$ is the training set with size $D$, $y$ is the golden answer for input $x$, $R(\tau, y)$ is the reward function, and $\pi_\text{ref}$ is a reference policy. We use GRPO (Shao et al., 2024) as our base RL algorithm.

### 3.2 AdaSearch

**Stage 1: Problem solving.**  We focus on incentivizing problem-solving capabilities for the policy LLM. Concretely, we train the policy to utilize (1) parametric knowledge and (2) search call for problem solving. We design two different system prompts: (1) parametric-knowledge prompt $s_\text{param}$, which requires the policy to answer using only its internal knowledge, and (2) search prompt $s_\text{search}$, which permits the use of search tools. Full prompt details are provided in Appendix E.

The training procedure is depicted in Algorithm 1. For each training instance $(x, y)$ in the training batch, we augment the problem $x$ with both $s_\text{param}$ and $s_\text{search}$, and generate two groups of rollouts: $\mathcal{R}_\text{param} = \{\tau_\text{param}^1, \tau_\text{param}^2, \ldots, \tau_\text{param}^N\}$ and $\mathcal{R}_\text{search} = \{\tau_\text{search}^1, \tau_\text{search}^2, \ldots, \tau_\text{search}^M\}$.

---

[1]Huggingface: `PeterJinGo/SearchR1-nq_hotpotqa_train-qwen2.5-7b-em-ppo`

---

**Algorithm 1** ADASEARCH Training

---

**Require:** base policy LLM $\pi_{\theta_b}$; search engine $\mathcal{E}$; training set $\mathcal{D} = (x_i, y_i)$; prompts $s_{\text{param}}$, $s_{\text{search}}$, $s_{\text{decision}}$; string concatenation $[\cdot, \cdot]$; solve-rate threshold $\rho$.
1: Initialize policy $\pi_\theta \leftarrow \pi_{\theta_b}$
2: **for** iteration = 1 to $T$ **do**    $\triangleright$ Stage-1 Training Loop
3:    Sample a batch $\mathcal{D}_b \sim \mathcal{D}$
4:    $\mathcal{R} \leftarrow \{\}$; $\mathcal{A} \leftarrow \{\}$
5:    **for** each $(x_i, y_i) \sim \mathcal{D}_b$ **do**
6:       **for** $g \in \{\text{param}, \text{search}\}$ **do**    $\triangleright$ Rollout Phase
7:          **if** $g$ is param **then**
8:             Generate $\mathcal{R}_g^i = \{\tau_{\text{param}}^n\}_{n=1}^N$, where $\tau_{\text{param}}^n \sim \pi_\theta(\cdot \mid [s_{\text{param}}, x_i])$.
9:          **else**
10:             Generate $\mathcal{R}_g^i = \{\tau_{\text{search}}^m\}_{m=1}^M$, where $\tau_{\text{search}}^m \sim \pi_\theta(\cdot \mid [s_{\text{search}}, x_i]; \mathcal{E})$.
11:          Compute rewards for all $\tau \in \mathcal{R}_g^i$ with Eq 4.
12:          Compute group-wise advantages $\mathcal{A}_g^i$ for $\mathcal{R}_g^i$.
13:          $\mathcal{R} \leftarrow \mathcal{R} \cup \mathcal{R}_g^i$; $\mathcal{A} \leftarrow \mathcal{A} \cup \mathcal{A}_g^i$    $\triangleright$ Aggregate Rollouts & Advantages
14:    Update $\pi_\theta$ using GRPO with $\mathcal{R}$ and $\mathcal{A}$.
15: Generate dataset $\mathcal{D}_{\text{decision}}$ with $\pi_\theta$, $\mathcal{D}$, and $s_{\text{decision}}$    $\triangleright$ Detailed in section 3.2
16: Stage-2 Training: Update $\pi_\theta$ with $\mathcal{D}_{\text{deision}}$ using GRPO
17: **return** $\pi_\theta$

---

For each trajectory $\tau \in \mathcal{R}_{\text{param}} \cup \mathcal{R}_{\text{search}}$, we use regular expressions to extract the final answer $\hat{y}$ from $\tau$ and apply exact match $\mathbf{EM}(\hat{y}, y) \rightarrow \{\texttt{true}, \texttt{false}\}$, which checks whether the extracted answer is exactly the same as any of the gold answer after normalization, to check the correctness. In the following sections, we use $\mathbf{EM}$ to denote the verification procedure above for simplicity.

The reward design in this stage is simply the binary correctness reward. Formally,

$$R(\tau, y) = \begin{cases} 1.0 & \text{if } \mathbf{EM} = \texttt{true}, \\ 0 & \text{otherwise.} \end{cases} \tag{4}$$

Finally, we compute group-wise advantages as in Shao et al. (2024) and update policy with Eq 3.

**Stage 2: Decision making.** The goal of stage 2 is to incentivize the decision making of search invocations. Before training, we use the stage 1 policy $\pi_{\theta_1}$ and the parametric-knowledge prompt $s_{\text{param}}$ to generate pseudo labels. Specifically, for each training instance $(x, y)$, we sample $K$ responses from $\pi_{\theta_1}$ and compute the empirical solve rate $p$ using substring exact match ($\mathbf{SubEM}$), which checks whether any gold answer is a substring of the extracted answer after normalization:

$$p = \frac{1}{K} \sum_{k=1}^K \mathbb{1}\left[\mathbf{SubEM}(\hat{y}_k, y) = \texttt{true}\right], \tag{5}$$

where $\hat{y}_k$ is the final answer extracted from the $k$-th response, and $\mathbb{1}[\cdot]$ is the indicator function. We use $\mathbf{SubEM}$ instead of the stricter $\mathbf{EM}$ because the objective here is not to evaluate exact answer formatting, but to estimate whether the model can solve the problem using its parametric knowledge. $\mathbf{SubEM}$ provides a more tolerant measure that avoids penalizing semantically correct answers with minor formatting differences, making it better suited for generating decision-making supervision.

Then, we define a threshold $\rho$ and label instances with $p \geq \rho$ as solvable with parametric knowledge. We use the token $\texttt{yes}$ as a label for such instances and the token $\texttt{no}$ otherwise. With the pseudo labels, we craft the dataset $\mathcal{D}_{\text{decision}} = \{(x_i, \ell_i)\}_{i=1}^D$, where $\ell_i \in \{\texttt{yes}, \texttt{no}\}$, for stage-2 training.

We design a decision-making system prompt $s_{\text{decision}}$ that requires explicit reasoning before producing the final decision $\ell$, as detailed in Appendix E. Such explicit reasoning increases transparency and interpretability of the decision-making process, making it clearer why the agent chooses to invoke search or rely solely on parametric knowledge. The reward function in this stage remains a simple binary outcome reward, identical to that used in stage 1 (Equation 4).

### 3.3 ADASEARCH INFERENCE

We adopt a two-stage inference pipeline (Figure 2). In the first stage, the model is prompted with the system instruction $s_{\text{decision}}$ to decide whether it can answer the query using parametric knowledge through explicit reasoning. In the second stage, the model receives the corresponding problem-solving instruction: if it decides it can answer with parametric knowledge, we use $s_{\text{param}}$; otherwise, we use $s_{\text{search}}$. Note that the stage-1 history is not prepended in stage 2.

## 4 EXPERIMENTS

### 4.1 EXPERIMENT SETUP

**Models.** We conduct experiments on Qwen2.5 (Team, 2024) and Llama-3.2 (Grattafiori et al., 2024) to ensure generalizability. Due to compute constraints, our main experiments use the 3B Instruct variants, with scaling results presented in Section 4.3.

**Search environment.** Following Jin et al. (2025a), we use E5 (Wang et al., 2022) as the retriever over a 2018 Wikipedia dump, with the implementation from Griggs et al. (2025). For each query, we return the top-3 documents and limit the number of search calls to at most three.

**Baselines.** We consider three categories of baselines. For **prompting baselines**, we select Direct inference, Chain-of-Thought (CoT) Reasoning (Wei et al., 2022), and RAG (Lewis et al., 2020). For **outcome-RL baselines**, we apply GRPO (Shao et al., 2024) to train models on the parametric prompt $s_{\text{param}}$ (RL w/o search) and on the search prompt $s_{\text{search}}$ (akin to Search-R1). For **reward-shaping baselines**, we include IKEA (Huang et al. (2025b), Table 3) as a representative method, and also design two baselines. The first one is denoted as "Naive Shaping". Its reward function is defined as

$$R(\tau, y) = \begin{cases} 1.0 - \lambda \cdot (\text{\# search calls}) & \text{if } \mathbf{EM} = \texttt{true}, \\ 0 & \text{otherwise}, \end{cases} \tag{6}$$

which is conceptually identical to OTC (Wang et al., 2025) but easier to implement, assigning higher rewards to correct rollouts with fewer search calls. We set $\lambda = 0.05$.

In addition, we introduce another reward-shaping baseline designed to enforce prompt-level search decisions, which closely resembles the objective of ADASEARCH. We refer to this variant as "Awareness Shaping". Although it directly encourages knowledge-aware search behavior, it still underperforms compared to ADASEARCH (Table 2). Its reward function is defined as

$$R(\tau, y) = \mathbb{1}\left[\mathbf{EM} = \texttt{true}\right] + \alpha \cdot \begin{cases} \mathbb{1}\left[\tau \text{ has search}\right] & \text{if } p < \rho, \\ \mathbb{1}\left[\tau \text{ has no search}\right] & \text{otherwise}, \end{cases} \tag{7}$$

where $p$ is the empirical solve rate defined in Eq 5, $\rho$ is the solve-rate threshold, and $\alpha$ a hyper-parameter controlling the bonus for correct self-knowledge awareness. This baseline encourages awareness by rewarding the model for making the appropriate search decision. In our experiments, we set both $\alpha$ and $\rho$ to 0.5. We estimate $p$ for each sample using the base policy before RL training. Finally, the prompts used for each baseline are detailed in Appendix E.

**Training.** To ensure fair comparison across training-based baselines, we construct a difficulty-balanced training set of 8,192 samples following the procedure of Huang et al. (2025b). We estimate the solve rate of each problem using the base policy and split the data into easy and hard subsets using a threshold $\rho = 0.5$, then sample equal-sized portions from each subset. A validation set is constructed using the same procedure and used for checkpoint selection. Full details for dataset construction are presented in Appendix B.1. For ADASEARCH stage 2, we use the same threshold $\rho = 0.5$. Full training details are provided in Appendix B.2.

**Evaluation.** Following Jin et al. (2025a), we use **EM** as the task metric, apply greedy decoding, and evaluate on standard QA benchmarks. For single-hop QA, we use Natural Questions (Kwiatkowski et al., 2019), TriviaQA (Joshi et al., 2017), and PopQA (Mallen et al., 2023). For multi-hop QA, we use HotpotQA (Yang et al., 2018), 2WikiMultihopQA (Ho et al., 2020),

Table 2: Results of different model across benchmarks. **EM** denotes exact match. $F1_{aware}$ denotes the self-knowledge awareness score (see Section 2). **Avg** is computed by averaging **EM** and $F1_{aware}$ over all benchmarks. The best score is shown in **bold** and the second best in underline.

| Method | General QA | | | | | | Multi-Hop QA | | | | | | | | Avg. | |
| --- | --- | --- | --- | --- | --- | --- | --- | --- | --- | --- | --- | --- | --- | --- | --- | --- |
| | NQ | | TQ | | PopQA | | HotpotQA | | 2Wiki | | MuSiQue | | Bamboogle | | | |
| | EM | $F1_{aware}$ | EM | $F1_{aware}$ | EM | $F1_{aware}$ | EM | $F1_{aware}$ | EM | $F1_{aware}$ | EM | $F1_{aware}$ | EM | $F1_{aware}$ | EM | $F1_{aware}$ |
| **Qwen2.5-3B-Instruct** | | | | | | | | | | | | | | | | |
| *Prompting Baselines* | | | | | | | | | | | | | | | | |
| Direct Answer | 8.1 | 8.1 | 24.1 | 24.1 | 7.8 | 7.8 | 13.2 | 13.2 | 19.5 | 19.5 | 1.4 | 1.4 | 5.6 | 5.6 | 11.4 | 11.4 |
| CoT | 14.1 | 14.1 | 38.2 | 38.2 | 13.5 | 13.5 | 17.3 | 17.3 | 22.8 | 22.8 | 3.9 | 3.9 | 24.0 | 24.0 | 19.1 | 19.1 |
| RAG | 31.7 | 0.0 | 52.7 | 0.0 | 38.1 | 0.0 | 24.6 | 0.0 | 20.1 | 0.0 | 4.4 | 0.0 | 8.0 | 0.0 | 25.7 | 0.0 |
| *Outcome RL Baselines* | | | | | | | | | | | | | | | | |
| RL w/o search | 21.7 | 21.7 | 44.9 | 44.9 | 17.3 | 17.3 | 20.5 | 20.5 | 28.0 | 28.0 | 5.9 | 5.9 | 23.2 | 23.2 | 23.1 | 23.1 |
| Search-R1 | **42.7** | 0.0 | **60.1** | 0.3 | 43.9 | 0.0 | **36.9** | 0.1 | 37.3 | 0.2 | **16.1** | 0.0 | **29.6** | 0.0 | **38.1** | 0.1 |
| *Reward-Shaping Baselines* | | | | | | | | | | | | | | | | |
| Naive Shaping | 37.8 | 41.6 | 54.8 | 66.9 | 42.2 | 55.3 | 29.8 | 49.3 | 33.1 | 58.4 | 9.9 | 15.8 | 24.8 | 43.8 | 33.2 | 47.3 |
| Awareness Shaping | 37.7 | 45.9 | 56.1 | 70.3 | 41.8 | 61.3 | 32.1 | 57.5 | 35.9 | 61.1 | 12.5 | 21.7 | 22.4 | 41.1 | 34.1 | 51.3 |
| IKEA | 41.4 | 37.7 | 59.5 | 62.4 | **44.2** | 55.2 | 32.0 | **58.1** | 35.0 | 60.2 | 11.5 | 14.2 | 23.2 | 46.0 | 35.3 | 47.7 |
| ADASEARCH | 37.9 | **49.0** | 56.8 | **71.7** | 42.8 | **58.8** | 33.4 | 57.5 | **38.5** | **65.5** | 14.4 | **25.2** | 28.0 | **50.0** | 36.0 | **54.0** |
| **Llama-3.2-3B-Instruct** | | | | | | | | | | | | | | | | |
| *Prompting Baselines* | | | | | | | | | | | | | | | | |
| Direct Answer | 18.8 | 18.8 | 42.3 | 42.3 | 16.4 | 16.4 | 12.5 | 12.5 | 16.2 | 16.2 | 3.0 | 3.0 | 8.8 | 8.8 | 16.9 | 16.9 |
| CoT | 25.0 | 25.0 | 45.6 | 45.6 | 15.8 | 15.8 | 16.3 | 16.3 | 11.3 | 11.3 | 4.5 | 4.5 | 35.2 | 35.2 | 22.0 | 22.0 |
| RAG | 29.4 | 0.0 | 53.0 | 0.0 | 35.7 | 0.0 | 20.6 | 0.0 | 8.3 | 0.0 | 4.1 | 0.0 | 12.8 | 0.0 | 23.4 | 0.0 |
| *Outcome RL Baselines* | | | | | | | | | | | | | | | | |
| RL w/o search | 38.2 | 38.2 | 54.6 | 54.6 | 23.1 | 23.1 | 23.7 | 23.7 | 26.7 | 26.7 | 6.7 | 6.7 | 34.4 | 34.4 | 29.6 | 29.6 |
| Search-R1 | **46.1** | 1.7 | **64.4** | 2.5 | **44.8** | 1.5 | **38.2** | 6.6 | 34.8 | 1.0 | **14.6** | 1.3 | **42.4** | 0.0 | **40.7** | 2.1 |
| *Reward-Shaping Baselines* | | | | | | | | | | | | | | | | |
| Naive Shaping | 41.2 | 57.6 | 59.7 | 75.3 | 39.3 | 56.3 | 31.2 | 52.6 | 33.5 | 54.8 | 11.5 | 21.1 | 36.8 | 55.9 | 36.2 | 53.4 |
| Awareness Shaping | 41.4 | 58.8 | 61.5 | 75.4 | 41.2 | 61.4 | 33.7 | 57.5 | 33.7 | 48.1 | 13.0 | 31.4 | 32.0 | 58.1 | 36.6 | 55.8 |
| IKEA | 42.2 | 58.6 | 62.0 | 76.2 | 41.1 | 59.9 | 32.3 | 58.2 | 32.9 | 51.3 | 11.2 | 31.9 | 40.8 | 63.1 | 37.5 | 57.0 |
| ADASEARCH | 43.5 | **62.7** | 62.9 | **79.2** | 40.4 | **62.3** | 34.4 | **59.2** | 36.2 | **60.1** | 13.2 | **36.5** | 42.4 | **64.2** | 39.0 | **60.6** |

MuSiQue (Trivedi et al., 2022), and Bamboogle (Press et al., 2023). We also report $F1_{aware}$ (Section 2) to measure self-knowledge awareness.

## 4.2 MAIN RESULTS

**Comparison against prompting and outcome-RL baselines.** Table 2 summarizes the results of prompting and outcome-RL baselines. In terms of task performance **EM**, Search-R1 achieves the best overall results across benchmarks, and the RAG baseline consistently outperforms all prompting-based methods. This pattern reflects the characteristics of these benchmarks, where retrieval generally offers an advantage over purely parametric reasoning; the performance gap between RL w/o Search and Search-R1 further supports this observation. Regarding self-knowledge awareness $F1_{aware}$, however, both the RAG baseline and Search-R1 obtain scores close to zero across benchmarks due to their always-search behavior. The results of Search-R1 highlight the deficiency of naive QA correctness rewards in balancing parametric and external knowledge. Non-search baselines achieve higher $F1_{aware}$, but because they never explicitly assess their own knowledge boundaries, their awareness remains suboptimal. In comparison, our ADASEARCH maintains competitive **EM** while substantially improving self-knowledge awareness across different model families, achieving 54% to 60% relative gain in $F1_{aware}$ over Search-R1. These results highlight that our method can effectively elicit self-knowledge awareness without compromising task performance.

**Comparison against reward-shaping baselines.** As shown in Table 2, our ADASEARCH consistently outperforms intricate reward-shaping baselines in both **EM** and $F1_{aware}$. First, Naive Shaping generally underperforms other baselines in **EM**, suggesting that naively penalizing tool usage might hurt performance. On the other hand, although Awareness Shaping generally shows improvements in both **EM** and $F1_{aware}$ over Naive Shaping, the results are still suboptimal. We suspect that this may be due to distribution shifts: we use the base policy before training to compute the empirical solve rate $p$, but as training proceeds, the solve rate may change as well. Training on such offline-generated labels may therefore introduce noise. Lastly, IKEA (Huang et al., 2025b) generally outperforms the other reward-shaping baselines in both **EM** and $F1_{aware}$ (except on Qwen2.5-3B-Instruct). Its reward design can be viewed as combining both Naive Shaping (e.g., penalizing tool usage on correct trajectories) and Awareness Shaping (e.g., adding bonuses on incorrect trajectories that involve search calls). However, ADASEARCH still consistently outperforms all reward-shaping

baselines across model families, especially on challenging multi-hop QA where search is more genuinely required. This suggests that reward shaping often over-optimizes for reducing tool calls and fails to adapt to query difficulty. In contrast, ADASEARCH strengthens self-knowledge awareness while achieving better task performance through explicit binary supervision.

### 4.3 ANALYSIS

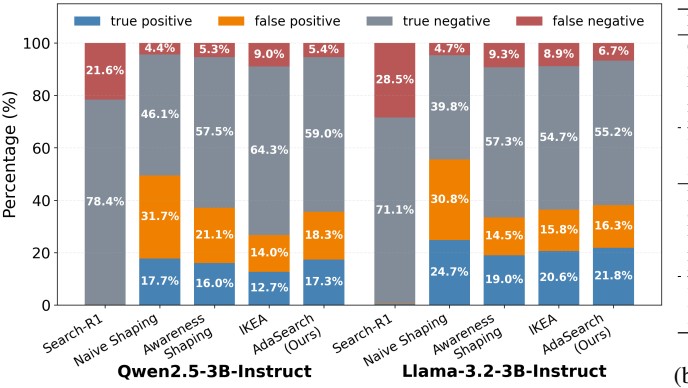

(a) Confusion Matrix

(b) **EM**, **F1**$_{\text{aware}}$, Precision, and Recall

| Method | EM | F1$_{\text{aware}}$ | Prec. | Rec. |
|---|---|---|---|---|
| **Qwen2.5-3B-Instruct** | | | | |
| Search-R1 | **38.1** | 0.1 | 39.0 | 0.0 |
| Naive Shaping | 33.2 | 47.3 | 36.1 | **76.8** |
| Awareness Shaping | 34.1 | 51.3 | 42.0 | 70.3 |
| IKEA | 35.3 | 47.6 | 45.0 | 53.4 |
| ADASEARCH | 36.0 | **54.0** | 45.1 | 68.8 |
| ADASEARCH-E2E | 35.2 | 51.3 | 44.1 | 63.9 |
| ADASEARCH-SFT | 37.0 | 51.1 | **45.9** | 59.5 |
| **Llama-3.2-3B-Instruct** | | | | |
| Search-R1 | **40.7** | 2.1 | 47.3 | 1.1 |
| Naive Shaping | 36.2 | 53.4 | 41.4 | **79.0** |
| Awareness Shaping | 36.6 | 55.8 | 52.1 | 60.5 |
| IKEA | 37.5 | 57.0 | 51.4 | 64.5 |
| ADASEARCH | 39.0 | **60.6** | **53.0** | 71.4 |
| ADASEARCH-E2E | 37.7 | 58.6 | 48.1 | 75.9 |
| ADASEARCH-SFT | 36.6 | 55.6 | 46.6 | 69.4 |

Figure 3: Analysis on self-knowledge awareness. (a) Confusion matrix of different RL methods. (b) Averaged **EM**, **F1**$_{\text{aware}}$, precision (**Prec.**), and recall (**Rec.**) across benchmarks.

**ADASEARCH achieves better self-knowledge awareness.** Figure 3 presents the confusion matrices and precision–recall–F1 scores across different methods and model families. For Naive Shaping, we observe the lowest false negative rates but substantially higher false positive rates for both Qwen and Llama (Figure 3a). This pattern suggests that naively penalizing search usage causes the model to underuse search: it spuriously avoids invoking search even when its parametric knowledge is insufficient, thereby drifting away from adaptive behavior. In contrast, ADASEARCH achieves comparable true positive and false negative rates while significantly reducing the false positive rate.

For the other reward-shaping baselines, although they show more balanced false positive and false negative rates, our ADASEARCH still outperforms them in terms of both **F1**$_{\text{aware}}$ and true positive rate. A closer examination shows that ADASEARCH attains precision on par with Awareness Shaping and IKEA, while achieving noticeably higher recall (Figure 3b).

**Two-stage training vs joint optimization.** To validate the benefit of two-stage training over joint optimization, we implement an end-to-end variant, denoted as **ADASEARCH-E2E**, which jointly optimizes problem solving and self-knowledge awareness. This design enables on-the-fly label generation for self-awareness by using the empirical solve rate of parametric knowledge at each step to produce pseudo labels. Further details are provided in Appendix D.

The results are shown in Figure 3b. ADASEARCH-E2E consistently underperforms ADASEARCH on both **EM** and **F1**$_{\text{aware}}$, suggesting that the two-stage approach is more effective than joint training.

**RL vs SFT in stage-2 training.** A common approach for teaching models to express uncertainty is supervised fine-tuning (SFT) (Lin et al., 2022; Zhang et al., 2024). In stage 2, we implement an SFT variant, **ADASEARCH-SFT**. After computing solve rates as described in Section 3.2, we directly construct target responses: for problems with solve rate $p \geq \rho$, we assign "`<assessment>yes</assessment>`," and "`<assessment>no</assessment>`" otherwise, with $\rho = 0.5$ in our experiments. Additional details are provided in Appendix B.

The results are shown in Figure 3b. While ADASEARCH-SFT improves **F1**$_{\text{aware}}$ over Search-R1 and achieves slightly higher **EM** on Qwen, it still underperforms ADASEARCH on Llama across both metrics. A closer look reveals that the **F1**$_{\text{aware}}$ gap is especially large on challenging benchmarks such as MuSiQue (Trivedi et al., 2022) and Bamboogle (Press et al., 2023), echoing findings from Chu et al. (2025) that RL generalizes better than SFT to out-of-distribution tasks.

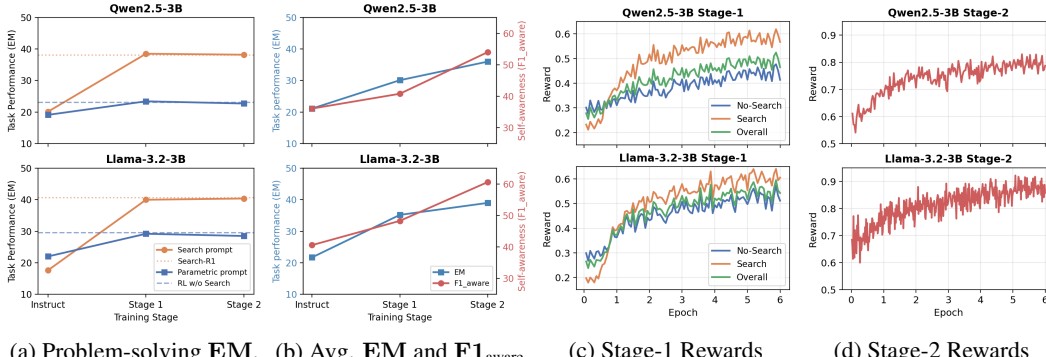

(a) Problem-solving **EM**.  (b) Avg. **EM** and **F1**$_{\text{aware}}$  (c) Stage-1 Rewards  (d) Stage-2 Rewards

Figure 4: Averaged performance and training rewards across stages. (a) Stage 1 substantially improves problem solving, and Stage 2 does not degrade it. (b) Both test **EM** and self-knowledge awareness (**F1**$_{\text{aware}}$) improve throughout training. (c) (d) Stage 1 and Stage 2 respectively incentivize problem-solving and self-knowledge awareness on the training set.

**Learning curves.** The learning curves are shown in Figure 4. In stage 1, the training reward increases steadily (Figure 4c), and the model's performance on both search-based and parametric problem solving matches the single-task baselines, Search-R1 and RL w/o Search (Figure 4a). In stage 2, the training reward also increases smoothly (Figure 4d), while self-knowledge awareness improves substantially and task performance continues to rise (Figure 4b). Moreover, Stage 2 does not degrade problem-solving ability under either prompt type (Figure 4a), consistent with recent findings that on-policy RL mitigates forgetting (Shenfeld et al., 2025; Chen et al., 2025a).

**Stage-2 labeling hyperparameters.** To understand the effect of labeling hyperparameters for stage-2 training, we conduct ablation studies on (1) the number of responses $K$ used to estimate the empirical solve rate in stage 2 and (2) the solve-rate threshold $\rho$ for stage-2 labeling.

The results are shown in Figure 5. For (1) (Figure 5a), increasing $K$ improves both **F1**$_{\text{aware}}$ and **EM** by providing a more reliable estimate of the solve rate. Notably, even $K = 1$ performs well (35.1% **EM** and 52.1% **F1**$_{\text{aware}}$), likely because RL sharpens output distributions(Yue et al., 2025; He et al., 2025), allowing a single sample to produce effective signals. For (2) (Figure 5b), $\rho$ has limited effect when $\rho \leq 0.5$, aside from a drop in **EM** at $\rho = 0.3$. As $\rho$ increases further, **F1**$_{\text{aware}}$ declines while **EM** rises, since higher $\rho$ encourages more `no` assessments, increasing search usage.

**Retriever choices.** To test the generalizability of ADASEARCH across different retrievers, we compare ADASEARCH when train and inference with E5-base (Wang et al., 2022) and BM25 (Robertson & Walker, 1994). As shown in Figure 5c, ADASEARCH continues to improve self-knowledge awareness even with the weaker BM25. While **EM** drops due to poorer retrieval, the self-awareness signal remains stable, indicating that ADASEARCH is robust to retriever choice.

**Model scaling.** To assess the robustness of our method across model scales, we train Qwen2.5-7B-Instruct and compare ADASEARCH with Search-R1 and Naive Shaping. As shown in Figure 5c, ADASEARCH boosts self-knowledge awareness on Qwen2.5-7B-Instruct by roughly 60% in **F1**$_{\text{aware}}$

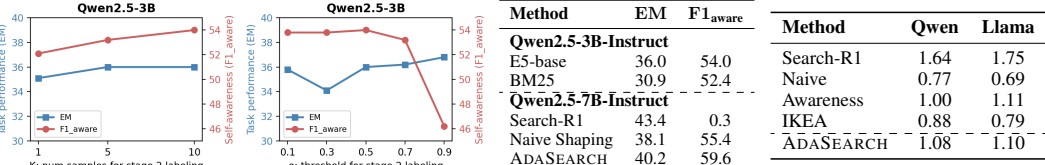

(a) Number of samples $K$ (b) Solve rate threshold $\rho$ (c) Retrievers & model size (d) Average number of search

Figure 5: Ablations of ADASEARCH.

compared to Search-R1. Relative to Naive Shaping, ADASEARCH improves both **EM** (+2.1%) and **F1**$_{aware}$ (+4.2%). These results indicate that ADASEARCH generalizes well to larger model scales.

**Average number of searches.** We report the average search calls across all benchmarks. ADASEARCH reduces unnecessary search usage by 34–38% compared to Search-R1, and its frequency is similar to Awareness Shaping since both rely on self-knowledge for prompt-level decision making. ADASEARCH uses slightly more searches than Naive Shaping and IKEA, which is expected because it does not directly optimize for minimizing tool calls.

## 5 RELATED WORK

**Retrieval-augmented generation.** Retrieval-augmented generation (RAG) has become a widely adopted paradigm for equipping LLMs with external knowledge (Shuster et al., 2021; Ram et al., 2023; Jiang et al., 2023; Asai et al., 2024; Wei et al., 2024). By integrating retrieval into the generation process, RAG has demonstrated strong potential to mitigate hallucinations and improve output accuracy across a range of real-world applications (Jin et al., 2025b; Lu et al., 2022; Tan et al., 2024; Xiong et al., 2025). Early approaches follow a static retrieve-and-read pipeline (Lewis et al., 2020; Guu et al., 2020; Izacard et al., 2023), which is effective for factoid queries but relatively limited for multi-step reasoning. Architectural variants such as RETRO (Borgeaud et al., 2022) deliver substantial gains but require model modifications and retraining, whereas in-context RAG (Ram et al., 2023) simply prepends retrieved passages to the prompt, enabling practical off-the-shelf use. More recent works emphasize reasoning-aware retrieval: IRCoT (Trivedi et al., 2023) interleaves Chain-of-Thought reasoning (Wei et al., 2022) with retrieval to improve multi-hop question answering, Adaptive-RAG (Jeong et al., 2024) adapts retrieval strategies to query complexity, and Deep-RAG (Guan et al., 2025) formulates retrieval as a step-wise decision process to balance parametric knowledge with external evidence. Our work shares a similar spirit but focuses on developing search agents via RL to adaptively decide whether to search through explicit reasoning.

**Reinforcement learning for search agents.** Reinforcement Learning (RL) has emerged as a powerful paradigm for augmenting LLMs with the ability to invoke external tools during reasoning, addressing tasks that require information beyond the model's internal knowledge (Guo et al., 2025; Jaech et al., 2024; Li et al., 2025). By treating search call as tools, LLMs are trained to act as search agents, interleaving reasoning with search in a multi-turn, interactive fashion. Representative methods include Search-R1 (Jin et al., 2025a) and ReSearch (Chen et al., 2025b). Following this line of work, recent studies have focused on explicitly optimizing for search adaptivity and efficiency. Huang et al. (2025b) train agents to delineate knowledge boundaries and synergize internal and external knowledge; Wang et al. (2025) seek to enhance tool productivity by reducing redundant calls through optimal tool call-controlled policy. However, these approaches rely on intricate reward shaping around search-call counts, requiring substantial manual tuning and trial-and-error. They also face ambiguous credit assignment, where agents exploit reward signals by superficially lowering call frequency instead of genuinely improving search behavior. Moreover, whether to invoke search remains an implicit decision, limiting transparency and hindering deployment in real-world, high-stakes settings. In contrast, we adopt a simple outcome-based RL framework that separates search decision making from problem solving, avoiding complex reward engineering while providing clearer learning signals. At inference time, ADASEARCH offers explicit decision rationales, improving transparency and making the agent's behavior easier to inspect.

## 6 CONCLUSION

In this work, we first conduct analysis and reveal that existing search agents often struggle to recognize the limits of their parametric knowledge, leading to leading to unnecessary or excessive search. Motivated by this observation, we propose ADASEARCH, a simple two-stage, outcome-driven RL framework that avoids complex reward engineering while improving both decision quality and task performance. ADASEARCH achieves the strongest self-knowledge awareness on both Qwen and Llama models, reduces unnecessary search calls, preserves task accuracy, and offers more transparent and interpretable decisions about when to search. These results demonstrate that ADASEARCH is an effective and generalizable approach for building adaptive, trustworthy search agents.

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

## A COMPARISON ON REWARD FUNCTIONS

We summarize the reward designs of different methods in Table 3. **OTC** (Wang et al., 2025) is designed to minimize search calls and maximize task performance. **IKEA** (Huang et al., 2025b) shares a similar spirit, but focus more on adaptivity by adding a knowledge-boundary bonus ($r_{kb}^-$) to encourage agents using search on difficult problems. **Naive Shaping** is conceptually similar to OTC but much easier to implement, aiming to encourage agents to use fewer search calls while obtaining the correct answer. **Awareness Shaping** aims to sharply penalize inconsistent transitions between no-search and search behaviors. In contrast, our ADASEARCH use a simple binary rewards to optimize both problem solving and decision making on whether to search.

## B IMPLEMENTATION DETAILS

### B.1 TRAINING DATASET CONSTRUCTION

To ensure a fair comparison across training-based baselines, we follow the procedure of Huang et al. (2025b) to construct a difficulty-balanced training set. Specifically, we use the base policy with the

Table 3: Comparison of different reward designs. Unlike prior methods that rely on complex reward engineering, such as OTC (Wang et al., 2025) and IKEA (Huang et al., 2025b), ADASEARCH uses a simple binary outcome reward to explicitly optimize both problem solving and decision making.

---

**Search-R1 (Jin et al., 2025a) & ADASEARCH (Ours)**

$$R(\tau, y) = \begin{cases} 1.0 & \text{if } \mathbf{EM} = \texttt{true}, \\ 0 & \text{otherwise}. \end{cases}$$

---

**OTC-GRPO (Wang et al., 2025)**

$$R(\tau, y) = \alpha \cdot R_{\text{tool}} \cdot \mathbb{1}\left[\mathbf{EM} = \texttt{true}\right], \text{ where } R_{\text{tool}} = \begin{cases} 1.0 & \text{if } f(m, n) = n = 0, \\ \cos\left(\frac{m\pi}{2m+c}\right) & \text{if } n = 0, \\ \sin\left(\frac{f(m,n)\pi}{2n}\right) & \text{otherwise}. \end{cases} \qquad f(m, n) = \begin{cases} 0 & \text{if } m = n = 0, \\ m & \text{if } n = 0, \\ \frac{2nm}{m+n} & \text{otherwise}. \end{cases}$$

$\alpha$: hyperparameter, $m$: # search calls in $\tau$, $n$: min # search calls for input $x$ during training.

---

**IKEA (Huang et al., 2025b)**

$$R(\tau, y) = \begin{cases} -1 & \text{if incorrect format}, \\ \mathbb{1}\left[\mathbf{EM} = \texttt{true}\right] + R_{\text{kb}} & \text{otherwise}. \end{cases} \qquad R_{\text{kb}} = \begin{cases} r_{\text{kb}^+}\left(1 - \frac{\text{RT}}{\text{RT}_{\max}}\right) & \text{if } \mathbf{EM} = \texttt{true}, \\ 0 & \text{if } \mathbf{EM} = \texttt{false} \ \& \ \text{RT} = 0, \\ r_{\text{kb}^-} & \text{otherwise}. \end{cases}$$

RT: # search calls; $\text{RT}_{\max}$: max # search allowed; $r_{\text{kb}^+}, r_{\text{kb}^-}$: hyperparameters.

---

**Naive Shaping**

$$R(\tau, y) = \begin{cases} 1.0 - \lambda \cdot (\# \text{ search calls}) & \text{if } \mathbf{EM} = \texttt{true}, \\ 0 & \text{otherwise}, \end{cases}$$

where $\lambda$ is the hyperparameter controlling penalties for search calls.

---

**Awareness Shaping**

$$R(\tau, y) = \mathbb{1}\left[\mathbf{EM} = \texttt{true}\right] + \alpha \cdot \begin{cases} \mathbb{1}\left[\tau \text{ has search}\right] & \text{if } p < \rho, \\ \mathbb{1}\left[\tau \text{ has no search}\right] & \text{otherwise}, \end{cases}$$

where $\alpha$: self-awareness bonus, $p$: solve rate for the input problem, and $\rho$: threshold for solve rate $p$.

---

system prompt $s_{\text{probe}}$ (Table 11), which encourages the model to answer directly without accessing additional information, to generate $K = 10$ responses for each problem in the training set of Jin et al. (2025a), drawn from Natural Questions (Kwiatkowski et al., 2019) and HotpotQA (Yang et al., 2018). We then compute solve rates using substring exact matching (**SubEM**) for verification. Using a threshold $\rho = 0.5$, we split the dataset into easy and hard subsets and sample 4,096 examples from each to form the final training set. The same procedure is used to construct a validation set of 2,048 examples, which is used for checkpoint selection during evaluation.

We create datasets for each model independently, and all methods evaluated on the same model use the same dataset to ensure fair comparison.

## B.2 TRAINING DETAILS

Most RL training experiments are conducted on a compute node equipped with 4 NVIDIA A100-80GB GPUs. The only exception is the outcome-based RL baselines for the 3B models, which are trained on a machine with 2 NVIDIA H100-94GB GPUs, as these baselines require relatively fewer computational resources. We use full parameter fine-tuning for all experiments. To optimize training efficiency, we adopted DeepSpeed with Zero3 offload, gradient checkpointing, FlashAttention-2, and bfloat16 mixed precision training.

We adopt LlamaFactory (Zheng et al., 2024) for SFT. We conduct hyperparameter search on batch size $\in \{16, 32\}$ and learning rate $\in \{1e\text{-}6, 5e\text{-}6\}$. We only train for one epoch as the training loss quickly converged to nearly 0. We select the checkpoint that maximizes the product of validation **EM** and $\mathbf{F1}_{\text{aware}}$.

For RL training, we use SkyRL (Griggs et al., 2025) as our base framework. Hyperparameters for RL baselines and for ADASEARCH are shown in Table 4 and Table 5, respectively. Most hyperparameters follow Jin et al. (2025a) and Wang et al. (2025). For reward-shaping baselines, we increase the group size to 15 to match the compute used in problem-solving training (ADASEARCH stage 1).

For the reward-function hyperparameters, we set $\lambda$ in Naive Shaping (Equation (6)) to 0.05; both $\alpha$ and $\rho$ in Awareness Shaping to 0.5; and $r_{\text{kb}}^+$ and $r_{\text{kb}}^-$ in IKEA (Table 3) to 0.2 and 0.05, following the original IKEA paper (Huang et al., 2025b). In addition, we analyze the reward-function hyper-

parameters for Naive Shaping and compare our training hyperparameters with those in the original IKEA implementation in Appendix C.2.

For ADASEARCH stage 2, we search over the combinations of (batch size, mini-batch size) in (128, 64), (256, 128). For Llama-3.2-3B-Instruct, we use the (128, 64) setting, while for the Qwen2.5 series models, we use (256, 128).

For efficient rollouts, we use vLLM (Kwon et al., 2023) and set the temperature to 1.0, top-p to 1.0, and top-k to –1, following Jin et al. (2025a).

For baselines that involve non-search generation (e.g., RL w/o Search, rollouts for $s_{param}$ in ADASEARCH stage 1, and ADASEARCH stage 2), we set the maximum number of turns to 2, with a maximum generation length of 500 tokens in the second turn. In most cases, the full trajectory of these baselines contains only one turn unless the model fails to follow the required output format (e.g., the answer is not wrapped in `<answer>...</answer>` or the assessment is not wrapped in `<assessment>...</assessment>`). Similar to Search-R1 (Jin et al., 2025a), which appends a nudge prompt (Table 20) when the model violates the required format, we design nudge prompts (Table 21 and 22) to enforce correct formatting. This formatting-correction procedure is applied to all reward-shaping baselines as well to ensure fairness.

We save checkpoints at the end of each epoch. For RL baselines, we select the checkpoint that achieves the highest validation reward. For ADASEARCH stage 1, we use the final checkpoint before collapse. For ADASEARCH stage 2, we select the checkpoint that maximizes the product of validation **EM** and **F1**$_{aware}$.

Table 4: Hyperparameters for RL baselines. $^*$: in the second turn, the maximum generation length is set to 500 (See Appendix B.2).

| Hyperparameter | Search-R1 | RL w/o Search | Reward-Shaping Baselines |
|---|---|---|---|
| (batch size, mini-batch size) | (512, 256) | (512, 256) | (512, 256) |
| learning rate | 1e-6 | 1e-6 | 1e-6 |
| epoch | 6 | 6 | 6 |
| group size | 5 | 10 | 15 |
| KL loss coefficient $\beta$ | 0.001 | 0.001 | 0.001 |
| PPO clip ratio | 0.2 | 0.2 | 0.2 |
| warm-up step ratio | 0.285 | 0.285 | 0.285 |
| max generation length | 500 | 2048$^*$ | 500 |
| max number of turns | 4 | 2 | 4 |
| max input length | 4096 | 3072 | 4096 |

Table 5: Hyperparameters for ADASEARCH. $^*$: in the second turn, the maximum generation length is set to 500 (See Appendix B.2).

| Hyperparameter | ADASEARCH stage 1 | ADASEARCH stage 2 |
|---|---|---|
| (batch size, mini-batch size) | (512, 256) | $\{(128, 64), (256, 218)\}$ |
| learning rate | 1e-6 | 1e-6 |
| epoch | 6 | 6 |
| group size | $(s_{search}, s_{param}) = (5, 10)$ | 5 |
| KL loss coefficient $\beta$ | 0.001 | 0.001 |
| PPO clip ratio | 0.2 | 0.2 |
| warm-up step ratio | 0.285 | 0.285 |
| max generation length | $(s_{search}, s_{param}) = (500, 2048^*)$ | 2048$^*$ |
| max number of turns | $(s_{search}, s_{param}) = (4, 2)$ | 2 |
| max input length | $(s_{search}, s_{param}) = (4096, 3072)$ | 3072 |

## B.3 INFERENCE DETAILS

We perform inference with vLLM (Kwon et al., 2023) and bfloat16 precision on the same compute nodes used for training. For each method, the maximum generation length, maximum number of turns, and maximum input length are set to match the rollout configuration used during training for consistency. Following Jin et al. (2025a), we use greedy decoding for evaluation.

## C ADDITIONAL ANALYSIS

In this section, we analyze the effects of hyperparameters across different methods and examine efficiency in terms of the number of search calls and the latency. All analyses are conducted on Qwen2.5-3B-Instruct.

### C.1 ADASEARCH STAGE 2: THRESHOLD $\rho$ VS NUMBER OF SEARCH CALLS

We analyze the effect of the solve-rate threshold $\rho$ on the number of search calls, with results shown in Table 6. As $\rho$ increases, the number of search calls rises, which is expected since a higher threshold encourages the model to invoke search more often. However, when $\rho$ becomes too large (e.g., 0.9), $F1_{\text{aware}}$ drops substantially. This suggests overuse of search, where the model chooses to search even when its parametric knowledge is sufficient, as reflected in the decrease in recall and the increase in precision.

Table 6: Analysis on the effect of $\rho$ for ADASEARCH stage 2 (Section 3.2).

| $\rho$ | EM | $F1_{\text{aware}}$ | Precision | Recall | Avg. Search |
|---|---|---|---|---|---|
| 0.1 | 35.8 | 53.8 | 44.5 | 69.3 | 0.99 |
| 0.3 | 34.1 | 53.8 | 43.4 | 72.5 | 1.00 |
| 0.5 | 36.0 | 54.0 | 45.1 | 68.8 | 1.08 |
| 0.7 | 36.2 | 53.2 | 48.9 | 59.6 | 1.12 |
| 0.9 | 36.8 | 46.2 | 55.9 | 41.0 | 1.21 |

### C.2 HYPERPARAMETERS OF REWARD-SHAPING BASELINES

We analyze the reward-function hyperparameter of Naive Shaping and Awareness Shaping, and the training hyperparameters of IKEA (Huang et al., 2025b) on Qwen2.5-3B-Instruct.

**Naive shaping.** We vary the value of $\lambda$ in Equation 6 over 0.05, 0.1, 0.25, and report **EM**, $F1_{\text{aware}}$, precision, recall, and the micro-averaged number of search calls. The results are shown in Table 7. As expected, the average number of search calls decreases as $\lambda$ increases, due to the stronger penalty on invoking search. $F1_{\text{aware}}$ increases, which can be attributed to the increase in recall, while precision remains relatively stable. Lastly, **EM** decreases with larger $\lambda$, possibly because the model underuses search, as reflected by the lower recall and reduced number of search calls.

Table 7: Analysis on the effect of $\lambda$ for Naive Shaping baseline (Eq 6).

| $\lambda$ | EM | $F1_{\text{aware}}$ | Precision | Recall | Avg. Search |
|---|---|---|---|---|---|
| 0.05 | 33.2 | 47.3 | 36.1 | 76.8 | 0.77 |
| 0.1 | 33.0 | 47.8 | 35.7 | 81.3 | 0.74 |
| 0.25 | 32.1 | 48.7 | 35.9 | 85.6 | 0.67 |

**Awareness shaping.** We vary the value of $\alpha$ in Equation 7 over 0.1, 0.5, 1.0 and report **EM**, $F1_{\text{aware}}$, precision, recall, and the micro-averaged number of search calls. The results are shown in Table 8. Increasing $\alpha$ improves $F1_{\text{aware}}$ and recall, as expected from assigning a larger bonus to correct decisions. In contrast, precision, average search-call counts, and task **EM** decrease. These trends suggest that the model may underuse search, reflected in higher recall but lower precision and fewer search calls.

Table 8: Analysis on the effect of $\alpha$ for Awareness Shaping baseline (Eq 7).

| $\alpha$ | EM | $F1_{aware}$ | Precision | Recall | Avg. Search |
|---|---|---|---|---|---|
| 0.1 | 36.2 | 49.5 | 44.7 | 57.5 | 1.12 |
| 0.5 | 34.1 | 51.3 | 42.0 | 70.3 | 1.00 |
| 1.0 | 33.8 | 52.5 | 41.6 | 76.4 | 0.98 |

**IKEA.** We compare our training hyperparameters for IKEA (Huang et al., 2025b) (Table 4) with those used in the original IKEA paper. In the original work, the authors train with both batch size and mini-batch size set to 256, a smaller learning rate of 5e-7, a much larger warm-up ratio of 0.75, a slightly larger group size of 16, and a total of 120 training steps (approximately 4 epochs). The performance comparison is shown in Table 9. We find that using the original hyperparameters results in IKEA being undertrained. Therefore, we adopt the hyperparameters in Table 4 for our experiments.

Table 9: Comparison of Hyperparameters for IKEA Huang et al. (2025b)

| Hyperparameter Setting | EM | $F1_{aware}$ | Precision | Recall | Avg. Search |
|---|---|---|---|---|---|
| Original (Huang et al., 2025b) | 32.3 | 12.4 | 42.0 | 8.0 | 1.06 |
| Ours (Table 4) | 35.3 | 47.6 | 45.0 | 59.5 | 0.88 |

### C.3 COMPARISON ON EFFICIENCY

In this section, we compare the efficiency of each method in terms of the average number of search calls and the average latency, as shown in Table 10. All averages are computed over the full set of testing samples. In addition, we estimate the total search cost on the evaluation benchmarks using the pricing of the Google Search API (5 USD per 1,000 queries)[2].

Regarding the average number of search calls, ADASEARCH substantially reduces unnecessary searches and costs compared to Search-R1, achieving a 34% reduction. ADASEARCH also incurs a similar number of search calls as Awareness Shaping, as both methods decide whether to invoke search on prompt-level based on self-knowledge. Compared to methods that explicitly minimize search calls (Naive Shaping and IKEA), ADASEARCH uses slightly more searches. This behavior is expected, since ADASEARCH does not directly optimize for minimizing tool calls. Instead, it focuses on avoiding unnecessary searches when parametric knowledge is sufficient, while still achieving a significant reduction relative to Search-R1.

In terms of average latency, although ADASEARCH introduces an additional inference stage that explicitly decides whether to invoke search, its latency remains lower than that of Search-R1 by 20%. These results demonstrate that ADASEARCH not only reduces the average number of searches (and thus API cost) but also decreases inference latency, while providing transparent and interpretable decision rationales that are valuable in real-world applications. Compared to reward-shaping baselines, ADASEARCH incurs a slightly higher latency, but it achieves higher task performance (**EM**) and demonstrates better discrimination between necessary and unnecessary search calls, as reflected by $F1_{aware}$ (Table 2).

Finally, our framework is orthogonal to reward-shaping methods. Additional reward terms can be incorporated to penalize undesirable search behaviors, such as issuing duplicate queries. We leave a detailed exploration of this direction to future work.

## D ADASEARCH-E2E

We unify the two stages in ADASEARCH to obtain an end-to-end variant, ADASEARCH-E2E. The overall training pipeline is shown in Algorithm 2. For each training instance $(x, y)$, we construct

---

[2]https://developers.google.com/custom-search/v1/overview#pricing

Table 10: Average number of searches, their estimated costs, and the average latency.

| Method | Avg. Search | Est. Costs | Avg. Latency |
|---|---|---|---|
| **Qwen2.5-3B-Instruct** | | | |
| Search-R1 | 1.64 | 424.3 | 0.111 |
| Naive Shaping | 0.77 | 198.4 | 0.059 |
| Awareness Shaping | 1.00 | 259.0 | 0.074 |
| IKEA | 0.88 | 228.6 | 0.066 |
| ADASEARCH | 1.08 | 279.2 | 0.089 |

---

**Algorithm 2** ADASEARCH-E2E Training

---

**Require:** base policy LLM $\pi_{\theta_b}$; search engine $\mathcal{E}$; training set $\mathcal{D} = (x_i, y_i)$; prompts $s_{\text{param}}$, $s_{\text{search}}$, $s_{\text{decision}}$; string concatenation $[\cdot, \cdot]$; solve-rate threshold $\rho$.

1: Initialize policy $\pi_\theta \leftarrow \pi_{\theta_b}$
    /* End-to-End Training */
2: **for** iteration = 1 to $T$ **do**
3:     Sample a batch $\mathcal{D}_b \sim \mathcal{D}$
4:     $\mathcal{R} \leftarrow \{\}; \mathcal{A} \leftarrow \{\}$
5:     **for** each $(x_i, y_i) \sim \mathcal{D}_b$ **do**
6:         **for** $g \in \{\text{param}, \text{search}, \text{decision}\}$ **do**         ▷ Rollout Phase
7:             **if** $g$ is param **then**
8:                 Generate $\mathcal{R}_g^i = \{\tau_{\text{param}}^n\}_{n=1}^N$, where $\tau_{\text{param}}^n \sim \pi_\theta(\cdot \mid [s_{\text{param}}, x_i])$.
9:             **else if** $g$ is decision **then**
10:                 Generate $\mathcal{R}_g^i = \{\tau_{\text{decision}}^k\}_{k=1}^K$, where $\tau_{\text{decision}}^k \sim \pi_\theta(\cdot \mid [s_{\text{decision}}, x_i])$.
11:             **else**
12:                 Generate $\mathcal{R}_g^i = \{\tau_{\text{search}}^m\}_{m=1}^M$, where $\tau_{\text{search}}^m \sim \pi_\theta(\cdot \mid [s_{\text{search}}, x_i]; \mathcal{E})$.
13:         Compute rewards for all $\tau \in \mathcal{R}_g^i$ with Eq 4.
14:         Compute group-wise advantages $\mathcal{A}_g^i$ for $\mathcal{R}_g^i$.
15:         $\mathcal{R} \leftarrow \mathcal{R} \cup \mathcal{R}_g^i; \mathcal{A} \leftarrow \mathcal{A} \cup \mathcal{A}_g^i$         ▷ Aggregate rollouts & advantages
16:     Update $\pi_\theta$ using GRPO with $\mathcal{R}$ and $\mathcal{A}$.
    **return** $\pi_\theta$

---

three variants of the input by applying the parametric knowledge prompt $s_{\text{param}}$, the search prompt $s_{\text{search}}$, and the decision-making prompt $s_{\text{decision}}$.

The pseudo labels for decision making are generated on the fly: for each training instance $(x, y)$, we estimate the empirical solve rate $p$ from $\mathcal{R}_{\text{param}}$ using **SubEM**, and assign pseudo labels according to whether $p$ exceeds a predefined threshold $\rho$. The reward function is simply the binary outcome reward, and the remaining optimization follows GRPO (Shao et al., 2024).

The hyperparameters largely follow those of our two-stage method (Table 5), except for the batch and mini-batch sizes. We search over $\{(256, 128), (512, 256)\}$ and use $(256, 128)$ for Qwen and $(512, 256)$ for Llama.

# E  PROMPT TEMPLATES

Table 11: Knowledge-probing system prompt for dataset construction.

---

**Knowledge-Probing System Prompt $s_{probing}$**

You are a helpful assistant. Your goal is to answer questions using step-by-step reasoning. Use only your internal knowledge to recall and connect relevant information. Do NOT assume access to external tools or documents. Provide the final answer only after completing your reasoning, following the output instructions below.

## Output Instructions
1. Wrap your final answer within `<answer>` and `</answer>` tags. For example: `<answer>`Wilhelm Conrad Röntgen`</answer>`.
2. Only include the direct final answer within these tags; place all explanations and reasoning outside.

---

Table 12: System prompt for search, adapt from Jin et al. (2025a)

.

---

**Search System Prompt $s_{search}$**

Answer the given question. You must conduct reasoning every time you get new information. After reasoning, if you find you lack some knowledge, you can call a search engine by `<search>`query`</search>`, and it will return the top searched results between `<output>` and `</output>`. You can search as many times as you want. If you find no further external knowledge needed, you can directly provide the answer inside `<answer>` and `</answer>` without detailed illustrations. For example, `<answer>`xxx`</answer>`.

---

Table 13: Parametric-knowledge system prompt. This prompt is used for CoT and RL w/o Search baselines as well.

---

**Parametric-knowledge System Prompt $s_{param}$**

Answer the given question. Always conduct and show your step-by-step reasoning first, then give the final answer. When providing the answer, wrap it inside `<answer>` and `</answer>` without detailed illustrations. For example, `<answer>`xxx`</answer>`.

---

Table 14: Decision-making system prompt.

---

**Decision-Making System Prompt $s_{decision}$**

Your goal is to assess whether your knowledge is sufficient to answer the given question. Think step-by-step and then output the final assessment by following the instructions below:

## Output Instructions
1. Always output your step-by-step reasoning first.
2. After your reasoning, output a single assessment token wrapped within `<assessment>` ... `</assessment>` tags.
    – If you are confident you can answer the question directly, output: `<assessment>`yes`</assessment>`
    – Otherwise, output: `<assessment>`no`</assessment>`
3. Only include the final assessment token within these tags; place all reasoning outside the tags.

---

Table 15: System prompt for reward-shaping baselines (Naive Shaping, Awareness Shaping, and IKEA), adapt from Wang et al. (2025)

| System Prompt for Reward-shaping Baselines |
|---|
| Answer the given question. You must conduct reasoning every time you get new information. After reasoning, if you find you lack some knowledge, you can call a search engine by `<search>`query`</search>`, and it will return the top searched results between `<output>` and `</output>`. You need to make every search call count and gain helpful results. If you find no further external knowledge needed, you can directly provide the answer inside `<answer>` and `</answer>` without detailed illustrations. For example, `<answer>`xxx`</answer>`. |

Table 16: System prompt for direct answer baseline.

| System Prompt for Direct Answer Baseline |
|---|
| Answer the given question. When providing the answer, wrap it inside `<answer>` and `</answer>` without detailed illustrations. For example, `<answer>`xxx`</answer>`. |

Table 17: System prompt for RAG baseline.

| System Prompt for RAG Baseline |
|---|
| Answer the given question based on the provided documents. When providing the answer, wrap it inside `<answer>` and `</answer>` without detailed illustrations. For example, `<answer>`xxx`</answer>`. |

Table 18: Default user prompt template, where `question` will be replaced with the actual input question.

| Default User Prompt Template |
|---|
| Question: {`question`} |

Table 19: RAG User prompt template, where `document_i` and `question` will be replaced with the retrieved document and the actual input question, respectively.

| RAG User Prompt Template |
|---|
| Doc 1: {`document_1`}
Doc 2: {`document_2`}
Doc 3: {`document_3`}
Question: {`question`} |

Table 20: Retry prompt used when the model produces an invalid action under the search prompt $s_{\text{search}}$.

> **Retry Prompt (Search)**
>
> My previous action is invalid. If I want to search, I should put the query between `<search>` and `</search>`. If I want to give the final answer, I should put the answer between `<answer>` and `</answer>`. Let me try again.

Table 21: Retry prompt used when the model fails to wrap the answer in `<answer>...</answer>` under the parametric-knowledge prompt $s_{\text{param}}$.

> **Retry Prompt (Parametric-knowledge)**
>
> I should put the answer between `<answer>` and `</answer>`.

Table 22: Retry prompt used when the model produces a search call under the parametric-knowledge prompt $s_{\text{param}}$.

> **Retry Prompt (Parametric-knowledge; When search is invoked)**
>
> My previous action is invalid. I should rely on my own step-by-step reasoning to solve the problem and then put the answer between `<answer>` and `</answer>`. Let me try again.

## F  CASE STUDIES

### F.1  CASE 1: PARAMETRIC KNOWLEDGE IS SUFFICIENT

Table 23: Input question. The answer is "Christian Louboutin".

> **Input Question (Answer: Christian Louboutin)**
>
> Which designer is famous for his/her red-soled footwear?

Table 24: ADASEARCH stage-1 output. The model explicitly evaluates its own knowledge and concludes that it is sufficient.

> **ADASEARCH Stage-1 Output (Answer: Christian Louboutin)**
>
> Step 1: I need to think of a designer known for red-soled footwear.
> Step 2: One famous designer known for red-soled footwear is Christian Louboutin.
> Step 3: Christian Louboutin is a French fashion designer who is renowned for his distinctive red-soled high-heeled shoes.
> Step 4: Based on this information, I can answer the question.
> `<assessment>`yes`</assessment>`

Table 25: ADASEARCH stage-2 output. The model correctly answer the question using parametric knowledge.

| ADASEARCH Stage-2 Output (Answer: Christian Louboutin) |
|---|
| To determine which designer is famous for his/her red-soled footwear, we need to consider some of the most well-known designers and their iconic designs.

1. Christian Louboutin is a renowned French fashion designer who is famous for creating red-soled high-heeled shoes.
2. Other designers like Jimmy Choo and Alexander McQueen have also designed red-soled shoes, but Christian Louboutin is particularly known for this specific style.

Given this information, the designer famous for his/her red-soled footwear is Christian Louboutin.

`<answer>`Christian Louboutin`</answer>` |

Table 26: Search-R1 output. The agent overuses search on problems where its parametric knowledge is sufficient.

| Search-R1 Output (Answer: Christian Louboutin) |
|---|
| To find out which designer is famous for their red-soled footwear, I will need to search for information about notable footwear designers and their signature styles.

`<search>`Which designer is famous for red-soled footwear`</search>`
`<output>`
Doc 1: "Christian Louboutin"
Christian Louboutin Christian Louboutin (; born 7 January 1964) is a French fashion designer whose high-end stiletto footwear incorporates shiny, red-lacquered soles that have become his signature. Initially a freelance designer for fashion houses, he started his own shoe salon in Paris, with his shoes finding favor with celebrity clientele. He has partnered with other organizations for creative projects including limited edition pieces, gallery exhibits, and even a custom bar. His company has since branched out into men's footwear, handbags, fragrances and makeup. Louboutin was born and raised in Paris's 12th arrondissement. He was the only son of Roger, a
Doc 2: "Christian Louboutin"
tempered to 28,000 Mexican pesos (1471.89 dollars). Christian Louboutin Christian Louboutin (; born 7 January 1964) is a French fashion designer whose high-end stiletto footwear incorporates shiny, red-lacquered soles that have become his signature. Initially a freelance designer for fashion houses, he started his own shoe salon in Paris, with his shoes finding favor with celebrity clientele. He has partnered with other organizations for creative projects including limited edition pieces, gallery exhibits, and even a custom bar. His company has since branched out into men's footwear, handbags, fragrances and makeup. Louboutin was born and raised in Paris's 12th arrondissement. He
Doc 3: "Christian Louboutin"
on his high heel shoes, commonly referred to as ""sammy red soled shoes"". His signature red soles were first created in 1993. ""My sketches were not reproduced exactly as I had designed them and I couldn't figure out why. The two dimensional sketch was so powerful on paper, but when turned into a three-dimensional object, it was somehow lacking energy. Frustrated after having tried different things to liven up the design, I spontaneously grabbed my assistant's red nail polish and painted the sole. I instantly knew that this would be a success!"". Christian Louboutin's red-bottom colour code is Pantone 18-1663
`</output>`

From the search results, it is clear that Christian Louboutin is famous for his red-soled footwear.

`<answer>`Christian Louboutin`</answer>` |

Table 27: Naive Shaping output. The model produces an incorrect answer without explicit self-knowledge assessment.

---

**Naive Shaping Output (Answer: Christian Louboutin)**

To determine which designer is famous for their red-soled footwear, let's consider some prominent designers and their notable styles.

One of the most famous designers known for red-soled footwear is Alexander McQueen. He was known for his bold and innovative designs, including red-soled shoes that became iconic in his collections.
Given this information, the designer famous for his/her red-soled footwear is Alexander McQueen.

`<answer>`Alexander McQueen`</answer>`

---

## F.2    CASE 2: PARAMETRIC KNOWLEDGE IS INSUFFICIENT

Table 28: Input question. The answer is "Bardney".

---

**Input Question (Answer: Bardney)**

Southrey is a village in the civil parish of which village in the West Lindsey district of Lincolnshire, England?

---

Table 29: ADASEARCH stage-1 output. After self-assessment, the model is not confident in answering using parametric knowledge.

---

**ADASEARCH Stage-1 Output (Answer: Bardney)**

Step 1: Identify the key information in the question. The question asks about a civil parish that contains the village of Southrey.
Step 2: Recall that a civil parish is a territorial unit of local government in England, and it is often part of a larger administrative division such as a district.
Step 3: Think about the West Lindsey district in Lincolnshire, England. I need to find which village in this district contains Southrey.
Step 4: Without specific knowledge of the exact location, I would need to search for the civil parish that includes Southrey. However, I don't have immediate access to this information.
Step 5: Given the lack of specific knowledge about the exact civil parish containing Southrey, I cannot answer the question directly.
`<assessment>`no`</assessment>`

---

Table 30: ADASEARCH stage-2 output. The model invokes search and produces a correct answer.

---

**ADASEARCH Stage-2 Output (Answer: Bardney)**

To determine the village in the West Lindsey district of Lincolnshire, England, that is the civil parish of Southrey, I will need to search for this information.

<search>Village in the West Lindsey district of Lincolnshire that is the civil parish of Southrey</search>
<output>
Doc 1: Bardney
consists of crafts, trade and community stands, funfair rides, bar and BBQ, sports, children's races, car boot sales, classic vehicles, and an Exemption Dog Show. Originally part of the Wraggoe Wapentake, and of the Lincoln poor law union in the Parts of Lindsey, the parish is now part of West Lindsey District, and Lincolnshire County Council. The civil parish has become part of a group that consists of Bardney, Apley and Stainfield parishes. The villages of Southrey and Kingthorpe are also included. The 2011 electoral arrangements are: Bardney lies between 7 and 17 metres above sea level, on the edge
Doc 2: Snelland
Snelland Snelland is a village and civil parish in the West Lindsey district of Lincolnshire, England It is situated approximately north-east from the city and county town of Lincoln and about from the town of Market Rasen. The civil parish includes the hamlet of Swinthorpe. The population at the 2011 census was included in the civil parish of Friesthorpe. Snelland is listed in ""Domesday Book"" of 1086 as ""Sneleslunt"", with seven households. The nearby deserted medieval village of Swinthorpe is listed as ""Sonetorp"", with three households. There is evidence that a Cistercian grange existed at Swinthorpe, attached to Kirkstead Abbey.
Doc 3: "South Thoresby"
South Thoresby South Thoresby is a village and civil parish in the East Lindsey district of Lincolnshire, England. It is situated north-east from Horncastle and south-east from Louth. South Thoresby is about east from the A16 in the Lincolnshire Wolds. The civil parish extends much further to the south, over the A16 and to include the hamlets of Calceby (a former medieval village) and Driby, extending southwards to border with Langton by Spilsby. To the south-west is Haugh. It lies close to the northern boundary with Aby with Greenfield. The parish council administers Swaby, South Thoresby and Haugh, although these
</output>

From the search results, it is clear that Southrey is a village in the civil parish of Bardney. Therefore, the village in the West Lindsey district of Lincolnshire, England, that is the civil parish of Southrey is Bardney.

<answer>Bardney</answer>

---

Table 31: Naive Shaping output. The model produces an incorrect answer without explicit self-knowledge assessment.

---

**Naive Shaping Output (Answer: Bardney)**

To determine the village in the West Lindsey district of Lincolnshire, England, that is the civil parish of Southrey, we need to find information about the civil parishes and their boundaries.

Southrey is a village in the civil parish of Little Sleaford. Little Sleaford is a village located in the West Lindsey district of Lincolnshire, England.

Therefore, the civil parish of Southrey is Little Sleaford.

<answer>Little Sleaford</answer>

---

### F.3 CASE 3: ADASEARCH UNDERUSE SEARCH

Table 32: Input question. The answer is "Mary Elizabeth Braddon".

| Input Question (Answer: Mary Elizabeth Braddon) |
| --- |
| Who is the author of Lady Audley's Secret? |

Table 33: ADASEARCH stage-1 output. The model holds a false belief, which leads to an incorrect self-assessment.

| ADASEARCH Stage-1 Output (Answer: Mary Elizabeth Braddon) |
| --- |
| Step 1: I need to recall the author of the novel "Lady Audley's Secret."
Step 2: I know that "Lady Audley's Secret" is a novel.
Step 3: I remember that the author of "Lady Audley's Secret" is Mrs. Wilkie Collins.
Step 4: I can confirm that Mrs. Wilkie Collins is indeed the author of this novel.
Step 5: Based on my knowledge, I can answer the question directly.
<assessment>yes</assessment> |

Table 34: ADASEARCH stage-2 output. The model maintains its false belief, resulting in insufficient search.

| ADASEARCH Stage-2 Output (Answer: Mary Elizabeth Braddon) |
| --- |
| To determine the author of "Lady Audley's Secret," we need to consider the historical context and the well-known works of that era.

"Lady Audley's Secret" is a novel published in 1862. The author of this novel is Mrs. Wilkie Collins, a prominent English novelist of the Victorian era.

Therefore, the author of "Lady Audley's Secret" is <answer>Mrs. Wilkie Collins</answer> |

### F.4 CASE 4: ADASEARCH OVERUSE SEARCH

Table 35: Input question. The answer is "Politician".

| Input Question (Answer: Politician) |
| --- |
| What is Javier Alva Orlandini's occupation? |

Table 36: ADASEARCH stage-1 output. The model is not confident in its self-knowledge and therefore abstains.

---

**ADASEARCH Stage-1 Output (Answer: Politician)**

Step 1: I need to determine Javier Alva Orlandini's occupation.
Step 2: I do not have any information about Javier Alva Orlandini in my knowledge base.
Step 3: Without specific information about this person, I cannot answer the question directly.
`<assessment>`no`</assessment>`

---

Table 37: ADASEARCH stage-2 output. The model produces a correct answer after search.

---

**ADASEARCH Stage-2 Output (Search) (Answer: Politician)**

To determine Javier Alva Orlandini's occupation, I will need to search for information about him. Let me start with a search.
`<search>`Javier Alva Orlandini occupation`</search>`
`<output>`
Doc 1: "Javier Alva Orlandini"
Javier Alva Orlandini Javier Alva Orlandini (born December 11, 1927) is a lawyer and a Peruvian politician. A prominent member of the political party Popular Action, he serves as the President of the party. He served as 2nd Vice President of Peru during the government of Fernando Belaúnde from 1980 to 1985.. Throughout his life he became a Deputy (1963-1968), Minister of Government and Police (1965-1966), Senator (1980-1985 and 1990-1992), President of the Senate (1981-1982) and Congressman (1995-2000). In 2002, he became a Magistrate of the Constitutional Court, at the same time being elected by Congress as the President of
Doc 2: "Luis Orlandini"
Luis Orlandini Luis Orlandini (born 1964) is a Chilean guitarist and professor at the University of Chile and the Pontifical Catholic University of Chile. Orlandini studied guitar with Ernesto Quezada at the University of Chile and then with Eliot Fisk at the School of Music of Cologne. In 1989, he obtained First Place at the ARD International Music Competition in Munich, thus initiating an international career that has led him to perform in various countries and record with different European labels. He performs mainly as a soloist, but has also played with groups of different natures of chamber music, one
Doc 3: "Javier Alva Orlandini"
the Court for a 3-year term. He was a candidate for the Presidency of the Republic in 1985, but got 4% of the popular vote, being defeated by Alan García. He later served as President of Peru's constitutional tribunal. He has written several books, including ""Responding to the Dictatorship"", ""Yesterday, Today and Tomorrow"", the ""Vicious Circle"", ""I Minister"", ""Word of Honor"" and ""Yes I swear"", and numerous legal articles. The audits, representative legislative, legal, academic and professional are of extensive quality and are recognized by the Peruvian people and their various institutions that have given many honorary degrees. Javier Alva
`</output>`

Based on the search results, Javier Alva Orlandini is a lawyer and a Peruvian politician. He served as the President of the party Popular Action and was the 2nd Vice President of Peru during the government of Fernando Belaúnde from 1980 to 1985.

`<answer>`Politician`</answer>`

---

Table 38: Model output under the parametric-knowledge system prompt $s_{\text{param}}$. The model is uncertain about its answer, even though the answer is correct.

---

**ADASEARCH Stage-2 Output (Parametric Knowledge) (Answer: Politician)**

To determine Javier Alva Orlandini's occupation, we need to look up information about him. Given that the question does not provide specific details, we can infer based on common knowledge or a search.

Javier Alva Orlandini is a Mexican politician who has held various positions in the Mexican government. Given his background, it is most likely that he is a politician.

However, without a specific source, we can't be certain. For the purpose of this question, we will assume his occupation is based on his known roles in Mexican politics.

`<answer>`Politician`</answer>`

---

## F.5 CASE 5: MULTI-HOP FAILURE OF ADASEARCH

Table 39: Input question. The answer is "1978".

---

**Input Question (Answer: 1978)**

The director of Panic 5 Bravo was born in what year?

---

Table 40: ADASEARCH stage-1 output. The model abstains because it lacks sufficient information.

---

**ADASEARCH Stage-1 Output (Answer: 1978)**

Step 1: To answer this question, I need to know the director of the movie "Panic 5 Bravo". However, there is no widely known movie or film titled "Panic 5 Bravo" that I am aware of. This suggests it might be a fictional or obscure title.
Step 2: Without knowing the director of this specific movie, I cannot determine the year of birth of the director.
Step 3: Since I do not have the necessary information to answer the question, I cannot provide a year of birth for the director.
`<assessment>`no`</assessment>`

---

Table 41: ADASEARCH stage-2 output (part 1). The model invokes search but makes a faulty inference (highlighted in red).

---

**ADASEARCH Stage-2 Output (Answer: 1978)**

To find out the year the director of "Panic 5 Bravo" was born, I will need to gather information about this film and its director. Let's start by searching for more details about the film "Panic 5 Bravo".

<search>The director of Panic 5 Bravo and the year of birth</search>

<output>

Doc 1: "Panic (2000 film)"

Panic (2000 film) Panic is a 2000 American crime film directed by Henry Bromell and starring William H. Macy, Neve Campbell, Tracey Ullman, John Ritter and Donald Sutherland. Despite not being a box office success, ""Panic"" received universal critical acclaim from critics, some of which called it one of the best movies of 2000. The performances of William H. Macy, John Ritter, and Donald Sutherland were hailed by critics. Alex (Macy), a sad-eyed mournful man, goes into psychotherapy disclosing that he is a hit man and also tells the doctor (Ritter) after a few sessions that he is attracted to

Doc 2: "Panic (2000 film)"

originality from the tone: quirky yet convincing, irreverent yet moral."" Panic (2000 film) Panic is a 2000 American crime film directed by Henry Bromell and starring William H. Macy, Neve Campbell, Tracey Ullman, John Ritter and Donald Sutherland. Despite not being a box office success, ""Panic"" received universal critical acclaim from critics, some of which called it one of the best movies of 2000. The performances of William H. Macy, John Ritter, and Donald Sutherland were hailed by critics. Alex (Macy), a sad-eyed mournful man, goes into psychotherapy disclosing that he is a hit man and also tells the doctor

Doc 3: "A Town Called Panic (film)"

A Town Called Panic (film) A Town Called Panic () is a 2009 internationally co-produced stop-motion animated adventure fantasy comedy family film directed by Stéphane Aubier and Vincent Patar and co-written by Aubier, Guillaume Malandrin, Patar, and Vincent Tavier. The film is based on the TV series of the same name and stars Aubier, Jeanne Balibar, Nicolas Buysse, Véronique Dumont, Bruce Ellison, Frédéric Jannin, Bouli Lanners, and Patar, among others. It premiered at the 2009 Cannes Film Festival and was the first stop-motion film to be screened at the festival. The film was released theatrically in Belgium on 17 June

</output>

From the search results, it seems "Panic 5 Bravo" is not a widely known film. However, the first result mentions a film called "Panic (2000 film)" directed by Henry Bromell. Given this information, I will now search for Henry Bromell's birth year.

---

Table 42: ADASEARCH stage-2 output. Continuing from part 1, the model uses the wrong director as the search query, which ultimately leads to an incorrect answer.

---

**ADASEARCH Stage-2 Output (Continued) (Answer: 1978)**

`<search>`Henry Bromell birth year`</search>`
`<output>`
Doc 1: "Henry Bromell"
Henry Bromell Alfred Henry Bromell (September 19, 1947 – March 18, 2013) was an American novelist, screenwriter, and director. Bromell joined the crew of NBC police drama """" in 1994. He served as a writer and co-executive producer for the show's third season. He contributed to writing seven episodes for the season. He was promoted to executive producer for the fourth season and wrote a further 17 episodes. He scaled back his involvement with the fifth season and became a consulting producer. He wrote a further two episodes before leaving the crew at the end of the season in 1997.
Doc 2: "Henry Bromell"
screenwriter and director Caroline Thompson. He then married writer Trish Soodik, who died of cancer in January 2009; they had a son, William. His third wife was Sarah. Bromell died March 18, 2013 of a heart attack, at UCLA Santa Monica hospital at age 65. Henry Bromell Alfred Henry Bromell (September 19, 1947 – March 18, 2013) was an American novelist, screenwriter, and director. Bromell joined the crew of NBC police drama """" in 1994. He served as a writer and co-executive producer for the show's third season. He contributed to writing seven episodes for the season. He was promoted
Doc 3: "Henry Bromell"
He contributed to a total of 26 episodes as a writer over three seasons with the series. He returned as a co-writer and co-executive producer for the feature-length follow-up ""Homicide: The Movie"" in 2000. He wrote and produced for many television series, including ""Chicago Hope"", ""Northern Exposure"", ""Homicide: Life on the Street"", ""Brotherhood"", ""Carnivàle"", and ""Rubicon"". He was a consulting producer, and later Executive Producer on the Showtime series ""Homeland"" at the time of his death and wrote four episodes: ""The Good Soldier"", ""Representative Brody"", ""Q&A"", and ""Broken Hearts"". He was awarded a for ""The Good Soldier"", and he was
`</output>`

From the search results, Henry Bromell was born in 1947. Therefore, the director of "Panic (2000 film)" was born in 1947.

`<answer>`1947`</answer>`

