# OpenReview forum: "Ada-Search: Balancing Parametric Knowledge and Search in Large Language Models via Reinforcement Learning"
_ICLR.cc/2026/Conference — Submitted to ICLR 2026_

### Official Review · Reviewer_TLUY · 2025-10-23

**Soundness:** 3
**Presentation:** 3
**Contribution:** 2
**Rating:** 4
**Confidence:** 4

**Summary:**

This paper addresses the challenge of training large language model (LLM) agents to use an external search tool only when necessary, rather than over-relying on search for every query. Overuse of search can incur unnecessary cost and risk exposure to irrelevant or malicious content. Previous approaches tried to curb excessive tool use by adding penalties for each search call, but this required complex reward engineering and led to ambiguous learning signals, agents might game the reward by avoiding calls even when needed.

**Strengths:**

The paper introduces a multi-task RL training framework (ADASEARCH) that explicitly optimizes both problem-solving and tool-use decision-making, instead of entangling them in a single reward.  It eliminates complex reward shaping – using a simple success/failure reward – yet achieves performance on par with or better than prior methods that used elaborate penalties.

**Weaknesses:**

* ADASEARCH  mainly re-frames existing tool-RL setups by partitioning prompts (decision vs. solving) and using simple outcome-only rewards to introduce a “should I search?” gate; this reads as an engineering refinement rather than a fundamentally new paradigm and thus feels incremental rather than a paradigm shift.

* ADASEARCH is consistently underperform by Search-R1 on EM across benchmarks. While the paper argues it reduces unnecessary tool calls, there is no accompanying cost/latency/token-use analysis (e.g., average tool calls per query, tokens per answer, wall-clock time, or $-cost) to demonstrate a concrete efficiency win that would justify the accuracy trade-off.

* Lack of qualitative case studies or error analysis. The paper would benefit from worked examples showing when the model correctly abstains from search, when it mistakenly avoids or overuses search, and typical multi-hop failure modes—with decision rationales and retrieved evidence.

* Limited evaluation breadth and realism. Testing on newer search benchmarks (e.g., BrowseComp, SimpleQA if applicable) and adding ablations with different retrievers and  responses numbers K would better demonstrate robustness.

**Questions:**

*  Please quantify average tool calls, latency, tokens, and $-cost.
*  Add a few worked examples showing correct abstentions, under-search, over-search, and typical multi-hop failures (with decision rationales).
*  Evaluate on newer browsing/search benchmarks (e.g., BrowseComp; SimpleQA if applicable) and vary retrievers，K and threshold  for robustness.

---

> ### Author Response · Authors · 2025-12-03
> **Response to Reviewer TLUY (1/2)**
>
> We sincerely thank the reviewer for the thoughtful response.
>
> ---
>
> **[W1]**
> > ADASEARCH mainly re-frames existing tool-RL setups...
>
> **[A1]**
>
> We emphasize that AdaSearch is not just a simple re-framing or an engineering refinement. Our contribution is to bring **trustworthiness** into search agents by introducing transparent decision making, explicit self-knowledge awareness, and auditable tool-use rationale. These properties are essential for real-world deployment and are largely missing from prior work, which focuses mainly on accuracy or tool-use reduction without addressing whether the model’s decisions can be understood or trusted.
>
> We summarize the following key importances and advantages of AdaSearch over prior works that solely focus on reducing the average number of tool calls:
> - **No complex reward engineering.** Unlike reward-shaping approaches that require carefully tuned penalties on search-call counts, AdaSearch uses a simple and principled outcome-based RL objective. This avoids the substantial engineering effort and the credit-assignment ambiguity introduced by handcrafted search penalties.
> - **Distinguishing necessary vs. unnecessary search.** Methods that only penalize the number of search calls conflate essential and redundant tool use. As reported in **Table 2**, reward-shaping baselines often underperform AdaSearch on harder multi-hop QA where search is more genuinely required, indicating their tendency to over-optimize for reducing tool calls potential failure to adapt to query difficulty relative to the model’s parametric knowledge.
> - **Explicit and interpretable decision making.** AdaSearch enables an interpretable and transparent reasoning process by providing explicit decision rationales for when search should be invoked (**Figure 1**; full content in **Appendix F.1**), which is fundamentally different from prior approaches in which tool use is driven implicitly through reward shaping. This transparency supports auditable diagnostics and offers promising avenues for improving model safety, such as through CoT monitoring [1][2].
>
> For these reasons, we believe that our formulation and the introduced F1 metric extend well beyond a simple re-framing and provide a valuable direction for future research on reliable search agents.
>
> ---
>
> **[Q1 & W2]**
> > Lack of cost/latency/token-use analysis to demonstrate a concrete efficiency win that would justify the accuracy trade-off.
>
> **[A2]**
>
> We report the average tool calls in **Figure 5d**, and the latency and estimated search costs in **Table 10**. Overall, our method **reduces search-call usage and costs by 34–38% and lowers latency by 20%** on Qwen2.5-3B-Instruct compared to Search-R1. These results demonstrate that our method effectively reduces unnecessary search calls together with the associated cost and latency.
>
> Compared to reward-shaping baselines, our method achieves higher task performance (EM; **Table 2**), better discriminates between necessary and unnecessary search calls (F1; **Table 2**), and provides a clear and interpretable rationale for why search is invoked (**Figure 1**), while introducing only a slightly higher number of search calls and a modest increase in latency.
>
> ---
>
> **[Q2 & W3]**
> > Lack of qualitative case studies or error analysis.
>
> **[A3]**
>
> We have added illustrative examples for each scenario in **Appendix F**. Under-search occurs when the model is overly confident in an incorrect belief; over-search occurs when the model triggers search even though its parametric knowledge would suffice; and multi-hop failures arise from misinterpreting retrieved evidence, as depicted in the reasoning trace after the first search results in the example.
>
> ---
>
> [1] Baker et al. "Monitoring reasoning models for misbehavior and the risks of promoting obfuscation." Anthropic 2025.
>
> [2] MacDiarmid et al. "Natural Emergent Misalignment From  Reward Hacking in Production RL. OpenAI 2025.

---

> ### Author Response · Authors · 2025-12-03
> **Response to Reviewer TLUY (2/2)**
>
> **[Q3-1 & W4]**
> > ... vary retrievers，K and threshold for robustness.
>
> **[A4]**
>
> Thank you for the suggestion. We have expanded our analysis to include several additional ablations. Specifically, we now report (1) the effect of varying the number of responses K used to estimate the solve rate p, (2) the impact of different thresholds \rho, and (3) results of training AdaSearch with BM25 rather than the default E5-base retriever. These results are provided in **Figure 5**.
>
> ---
>
> **[Q3-2 & W4]**
> >  Evaluate on newer browsing/search benchmarks (e.g., BrowseComp; SimpleQA if applicable)
>
> **[A5]**
>
> We note that our evaluation **exactly follows a rich line of prior works**  [1][2][3]. Our experiments span seven datasets covering both single-hop and multi-hop QA, providing broad coverage and reflecting the standard evaluation settings in related works.
>
> We agree that benchmarks such as BrowseComp and SimpleQA represent valuable testbeds. However, these tasks remain challenging for small language models. For example, BrowseComp requires interactive webpage browsing in addition to retrieval, which is beyond the capabilities of SLMs and is more suitable for evaluating deep-search or strong proprietary agents (and even then remains challenging [4]).
>
> For SimpleQA, Google recently introduced a cleaned version named SimpleQA Verified [5][6], explicitly stating that the benchmark is **“designed to be evaluated without any tools (i.e., search).”** Since our work focuses on learning when to invoke search tools, SimpleQA does not serve as a suitable evaluation for our setting.
>
> ---
>
> [1] Jin et al. Search-R1: Training LLMs to Reason and Leverage Search Engines with Reinforcement Learning. https://openreview.net/forum?id=Rwhi91ideu#discussion
>
> [2] Sun et al. ZeroSearch: Incentivize the Search Capability of LLMs without Searching. https://arxiv.org/abs/2505.04588
>
> [3] Wang et al. Acting Less is Reasoning More! Teaching Model to Act Efficiently. https://arxiv.org/abs/2504.14870v2
>
> [4] BrowseComp Leaderboard. https://www.kaggle.com/benchmarks/openai/browsecomp
>
> [5] SimpleQA Verified Leaderboard. https://www.kaggle.com/benchmarks/deepmind/simpleqa-verified
>
> [6] Haas et al. SimpleQA Verified: A Reliable Factuality Benchmark to Measure Parametric Knowledge. https://arxiv.org/abs/2509.07968v1

---

### Official Review · Reviewer_NEh8 · 2025-11-01

**Soundness:** 3
**Presentation:** 3
**Contribution:** 3
**Rating:** 6
**Confidence:** 3

**Summary:**

This submission presents an RL framework designed to help LLMs decide when to rely on their internal knowledge and when to use external search tools. The authors’ primary goal is to reduce the number of unnecessary search calls, while maintaining task performance.

To solve this problem, the authors introduce AdaSearch, an RL framework that trains LLMs to both solve tasks and make decisions about when to search. AdaSearch proceeds in two stages. In the first stage (problem solving), the model is trained with two prompts: one for parametric reasoning and one for search-based reasoning. The model is rewarded when it produces a correct final answer, as measured by an exact match with the ground truth. In the second stage (decision making), the model learns to decide whether to use search for a given question. Using “pseudo-labels” generated from the first stage’s solve rates, they train a binary classifier to predict whether external search is required. The authors also introduce an end-to-end variant that joints trains the problem-solving and decision-making modules.

The authors empirically evaluate their framework using the Qwen2.5 and Llama3 models, conducting experiments using question-answering benchmarks, comparing to various baselines. For the search component, they use E5 on a 2018 Wikipedia page dump.

The main results are presented in Table 2. The metrics the authors report are EM (task accuracy) and F1 score. Across the different tasks, AdaSearch performs comparable to or slightly better than the baselines.

**Strengths:**

AdaSearch is a clean RL framework for balancing an LLM’s internal knowledge with external search, without relying on hand-tuned reward functions. Their approach works by disentangling problem solving from deciding whether or not to search. I view the simplicity of their approach as a virtue and the main strength of this submission. Despite this simplicity, their method achieves comparable or slightly better performance across multiple QA benchmarks compared to more complicated baselines. Balancing search and internal knowledge is an important, and well-studied, problem. The paper writing is clear.

**Weaknesses:**

While I like the simplicity of this approach, the middling results when compared to the baselines make it unclear when the authors' approach should be used. I understand that previous approaches rely on reward engineering, but a discussion on when to use this approach versus other alternatives would be appreciated.

**Questions:**

n/a

---

> ### Author Response · Authors · 2025-12-03
> **Response to Reviewer NEh8**
>
> We sincerely thank the reviewer for the positive assessment and feedback.
>
> ---
>
> **[W1]**
> > a discussion on when to use this approach versus other alternatives would be appreciated.
>
> We agree that the choice between reward shaping and our approach depends on the deployment scenario.
> - Reward-shaping baselines can reduce average tool usage, but their behaviors are often difficult to audit. Because shaped rewards do not distinguish between “knowing the answer” and “being discouraged from calling a tool,” credit assignment becomes ambiguous, and the resulting reasoning process remains opaque to both users and model developers.
> - Our method differs fundamentally by introducing an **explicit decision step** in which the model must articulate whether its parametric knowledge is sufficient before invoking search. This provides transparent and auditable decision traces that reveal why the model chose to search or not to search, which is essential in high-stakes domains such as finance, medical QA, and legal compliance. In these domains, practitioners must understand whether failures arise from overconfidence, misinterpretation, or genuine knowledge gaps, which shaping-based methods cannot reveal.
>
> Overall, our method offers a more **reliable and interpretable** alternative to reward shaping: it elicits self-knowledge awareness through a simple, outcome-driven RL framework, avoids the trial-and-error tuning of reward-function hyperparameters, and provides explicit decision traces (**Figure 1**) that can be audited by users and developers.

---

### Official Review · Reviewer_jnAv · 2025-11-01

**Soundness:** 3
**Presentation:** 1
**Contribution:** 2
**Rating:** 4
**Confidence:** 4

**Summary:**

This paper proposes a reinforcement learning framework aimed at training large language models to decide when to rely on parametric knowledge versus invoking external search tools. The approach uses a two-stage training process with simple binary outcome rewards instead of complex reward shaping. The authors claim to reduce unnecessary search calls while maintaining task performance. Experiments are conducted on small 3B-parameter models (Qwen2.5 and Llama-3.2).

**Strengths:**

- Reducing unnecessary search calls in tool-augmented large language models is important both, economically and for system reliability.
- Empirical evaluation covers a diverse set of Q&A benchmarks. The decision-level F1 metric is a nice addition.
- The method significantly reduces the number of search calls.

**Weaknesses:**

- Missing ablation for the p-threshold and parameters of the reward-shaping based RL baselines.
- Benchmarks do not directly reflect an advantage of reduced function calls as Search-R1 consistently outperforms AdaSearch.
- Lack of hyperparameter settings (learning rate, ...), although promised to be in the appendix (page 6), limits reproducability.
- There is no evaluation of AdaSearch-E2E. The only comparison is made in Table 5. I suggest the authors focus on one of the methods or extend the analysis in the paper.
- The writing and formatting often seem rushed and incomplete. For example:
	- Table placement should be improved. Table 1 is mentioned on page 2 but appears on page 5.
	- Algorithm 2 is not mentioned at all.
	- Page 5: "[...] as detailed in Appendix X."
	- Hyperparameters promised in the appendix are missing.

**Questions:**

- Can you provide more evaluation results for AdaSearch-E2E?
- Could you provide ablation studies for naive shaping and AdaSearch? How does the threshold for p affect performance and tool use? How does it compare to different values of lambda?

---

> ### Author Response · Authors · 2025-12-03
> **Response to Reviewer jnAv**
>
> We sincerely appreciate the reviewer for the constructive feedback.
>
> ---
>
> **[W1]**
> > Missing ablation for the p-threshold and parameters of the reward-shaping based RL baselines.
>
> **[A1]**
>
> We have added ablations on the solve-rate threshold $\rho$ and the number of sampled responses $K$ used to estimate $p$, and report their effects on EM and F1 in **Figure 5a and 5b**. The impact of varying $\rho$ on tool usage is shown in **Table 6**, and the effects of different $\lambda$ values for Naive Shaping are reported in **Table 7**.
> In general, increasing the threshold $\rho$ slightly improves EM but raises tool usage, while overly large thresholds (for example, 0.9) lead to a sharp drop in F1 due to excessive search. For Naive Shaping, larger $\lambda$ values reduce tool calls but also lower EM, indicating that penalizing search too strongly can cause underuse of tools.
>
> ---
>
> **[W2]**
> > Benchmarks do not directly reflect an advantage of reduced function calls as Search-R1 consistently outperforms AdaSearch.
>
> **[A2]**
>
> We emphasize that our objective is not to surpass Search-R1 on task performance (EM), but to provide a more efficient, interpretable, and safety-aligned approach. We summarize the key advantages of AdaSearch below:
> - **Improved efficiency through reduced unnecessary search calls.** Overreliance on search calls introduces latency (from retrieval APIs and increased input length from retrieved documents) and inflates API cost. As shown in **Figure 5d**, AdaSearch reduces search calls and estimated costs by 34–38% while maintaining comparable EM.
> - **Mitigate safety and privacy risks by reducing external exposure.** Beyond efficiency, excessive search usage may expose the agent to noisy or malicious content and increase the risk of unintentional information leakage. By reducing unnecessary searches, AdaSearch lowers these risks while preserving task performance
> - **Explicit and interpretable decision-making.** Many real-world domains such as finance, healthcare, and privacy-sensitive applications require transparent and auditable tool-use decisions. AdaSearch provides explicit decision traces indicating why a search is or is not invoked (**Figure 1**), enabling auditable diagnostics and supporting safety techniques such as chain-of-thought monitoring [1][2]. In contrast, Reward-shaping baselines and Search-R1 do not offer such transparency.
>
> Overall, our intention is not to outperform Search-R1 on EM, but to avoid unnecessary search calls when the task can be solved by parametric knowledge, while improving safety, privacy, and interpretability through explicit decision-making rationales. These complementary benefits are not captured by EM alone but are essential for practical deployments.
>
> ---
>
> **[W3 & W5]**
> > Lack of hyperparameter settings (learning rate, ...); The writing and formatting often seem rushed and incomplete…
>
> **[A3]**
>
> Thank you for pointing these out. We have revised the manuscript to address the writing and formatting issues, and we now provide the hyperparameter settings for each method. Please refer to **Appendix B** for the full details.
>
> ---
>
> **[W4 & Q1]**
> > Can you provide more evaluation results for AdaSearch-E2E?
>
> **[A4]**
>
> We have added AdaSearch-E2E results using the E5 retriever in **Figure 3b** to enable a unified comparison with all baselines. We also report precision and recall in addition to EM and F1, and provide further discussion in **Section 4.3**. Overall, although E2E improves F1 relative to Search-R1, the two-stage method still performs better in both EM and F1. For this reason, we adopt the two-stage framework as our main approach and present the E2E results in the analysis section for completeness.
>
> ---
>
> **[Q2]**
> > Could you provide ablation studies for naive shaping and AdaSearch?
>
> **[A5]**
>
> Please refer to our response to **weakness 1 ([W1])**.
>
> ---
>
> [1] Baker et al. "Monitoring reasoning models for misbehavior and the risks of promoting obfuscation." Anthropic 2025.
>
> [2] MacDiarmid et al. "Natural Emergent Misalignment From  Reward Hacking in Production RL. OpenAI 2025.

---

### Official Review · Reviewer_rGmT · 2025-11-03

**Soundness:** 3
**Presentation:** 3
**Contribution:** 2
**Rating:** 2
**Confidence:** 4

**Summary:**

This paper studies the problem of training LLMs using reinforcement learning to efficiently use search calls to leverage retrieval tasks. In particular, the paper focuses on eliminating the use of search calls when the model's internal knowledge is sufficient to solve the task. To do so, the authors propose a straightforward inference method in which the model is first prompted to decide whether to use its own parametric knowledge (i.e. no search calls), or to use search in its response. The model is then trained using RL to improve all three of these abilities (deciding whether to use search or not, solving without search, and solving with search). The authors show that the resulting method is roughly on par with standard search + RL methods (e.g. Search-R1) in accuracy, but improves in terms of the proportion of tasks that don't utilize search (f1 score).

**Strengths:**

The paper tackles a clear problem and the proposed method is easy to understand and well-presented. The experiments are also well-done and almost all baselines are considered.

**Weaknesses:**

I found the paper to be primarily weak because of the lack of importance of the proposed problem setting. The paper focuses on reducing the proportion of tasks in which search is invoked, but I don't see a strong argument for why this is actually a useful distinction over prior work that reduces the average number of tool calls. For example, one could argue that not invoking tool calls on some prompts can reduce latency, but the proposed method requires an additional prompt to the LLM. Furthermore, because no existing method focuses on this problem setting, it's no surprise that the proposed metric of F1 score is better optimized by Ada-Search. Even then, there is a missing reward-shaping baseline that more sharply penalizes the transition between 0 and 1 tool calls.

**Questions:**

1. Why is SubEM necessary for the decision reward? If there's actually a big difference with EM, then using SubEM for everything may be better.
2. Stage 1 and 2 seems to be more of an ablation - I'd suggest keeping E2E as the main method, and introduce stage 1/2 only in the experiments section.
3. Beyond F1 score, are there results for average number of tool calls per problem for all methods?

---

> ### Author Response · Authors · 2025-12-03
> **Response to Reviewer rGmT (1/2)**
>
> We sincerely thank the reviewer for the thoughtful and constructive feedback.
>
> ---
>
> **[W1]**
> > Lack of importance of the proposed problem setting ... why this is actually a useful distinction over prior work that reduces the average number of tool calls ...
>
> **[A1]**
>
> We emphasize the following key importances and advantages of AdaSearch over prior works that solely focus on reducing the average number of tool calls:
>
> - **No complex reward engineering.** Unlike reward-shaping approaches that require carefully tuned penalties on search-call counts, AdaSearch uses a simple and principled outcome-based RL objective. This avoids the substantial engineering effort and the credit-assignment ambiguity introduced by handcrafted search penalties.
> - **Distinguishing necessary vs. unnecessary search.** Methods that only penalize the number of search calls conflate essential and redundant tool use. As reported in **Table 2**, reward-shaping baselines often underperform AdaSearch on harder multi-hop QA where search is more genuinely required, indicating their tendency to over-optimize for reducing tool calls potential failure to adapt to query difficulty relative to the model’s parametric knowledge.
> - **Explicit and interpretable decision making.** AdaSearch enables an interpretable and transparent reasoning process by providing explicit decision rationales for when search should be invoked (**Figure 1**; full content in **Appendix F.1**), which is fundamentally different from prior approaches in which tool use is driven implicitly through reward shaping. This transparency supports auditable diagnostics and offers promising avenues for improving model safety, such as through CoT monitoring [1][2].
>
> For these reasons, we believe that our problem setting  and the introduced F1 metric offer a meaningful direction for future work on reliable, trustworthy search agents.
>
> ---
>
> **[W2]**
> > For example, one could argue that not invoking tool calls on some prompts can reduce latency, but the proposed method requires an additional prompt to the LLM.
>
> **[A2]**
>
> We clarify that the additional prompt is introduced to provide an **explicit and interpretable decision-making step**. This transparency is essential in real-world, high-stakes domains such as finance and medical question answering, where users must be able to understand why the model decides to invoke or skip a tool call.
>
> Reducing unnecessary search calls also yields practical **safety and privacy** benefits, as it lowers exposure to noisy or malicious content and reduces the risk of unintended information leakage.
>
> To assess the efficiency impact, we computed the average end-to-end latency across all test samples using Qwen2.5-3B-Instruct. Our method still achieves **20% lower latency than Search-R1** (**Table 10**). This indicates that the small constant overhead of the additional prompt is outweighed by the efficiency gains from reducing unnecessary search calls. Compared to reward-shaping baselines, our method achieves higher task performance (EM; **Table 2**),  better discriminates between necessary and unnecessary search calls (F1; **Table 2**), and provides a clear and interpretable rationale for why search is invoked (**Figure 1**), while introducing only a slightly higher latency.
>
> ---
>
> **[W3]**
> > there is a missing reward-shaping baseline that more sharply penalizes the transition between 0 and 1 tool calls.
>
> **[A3]**
>
> We have added a reward-shaping baseline that applies a sharper penalty to the transition between 0/1 tool calls as described in **Equation 7**; the results are presented in **Table 2**. Even with this stronger shaping objective that directly targets self-knowledge awareness, AdaSearch still outperforms this baseline in both EM and F1. We also include ablation studies on the reward-function hyperparameters in **Table 8**.
>
> ---
>
> [1] Baker et al. "Monitoring reasoning models for misbehavior and the risks of promoting obfuscation." Anthropic 2025.
>
> [2] MacDiarmid et al. "Natural Emergent Misalignment From  Reward Hacking in Production RL. OpenAI 2025.

---

> ### Author Response · Authors · 2025-12-03
> **Response to Reviewer rGmT (2/2)**
>
> **[Q1]**
> > Why is SubEM necessary for the decision reward? If there's actually a big difference with EM, then using SubEM for everything may be better.
>
> **[A4]**
>
> We clarify that SubEM is not used as the decision reward. SubEM is applied only to estimate the empirical solve rate p when generating pseudo labels for stage 2 training, as described in **Section 3.2**. For training rewards and evaluation, we use EM for all methods, following the original Search-R1 setup, to ensure a fair and consistent comparison across baselines.
>
> ---
>
> **[Q2]**
> > Stage 1 and 2 seems to be more of an ablation - I'd suggest keeping E2E as the main method, and introduce stage 1/2 only in the experiments section.
>
> **[A5]**
>
> We initially considered presenting the E2E variant as the main method. After re-running the E2E experiments with the E5 retriever for consistency, we found that the two-stage formulation consistently outperforms E2E across both Qwen and Llama. This result is expected, as separating decision learning from answer generation stabilizes training and yields more reliable boundary-awareness behavior. For this reason, we keep the two-stage approach as our primary method, and include the E2E results (**Figure 3b**) and analysis (**Section 4.3**) for completeness.
>
> ---
>
> **[Q3]**
> > Beyond F1 score, are there results for average number of tool calls per problem for all methods?
>
> **[A6]**
>
> We have added the average number of tool calls for all methods in **Figure 5d**. Overall, our method reduces tool usage by 34-38% compared with Search-R1, demonstrating that it effectively avoids unnecessary search calls.

---

### Author Response · Authors · 2025-12-04
**General Response**

We sincerely thank all the reviewers for their constructive comments and suggestions. We are grateful for the recognition of our work in various aspects:
- The problem setting studied in this work is important (jnAv, NEh8) and easy to follow (rGmT, NEh8).
- The proposed method is simple and effective (rGmT, NEh8, TLUY).
- The empirical evaluation is comprehensive (rGmT,  jnAv, NEh8) and sound (jnAv, NEh8, TLUY).
- The presentation is clear (rGmT, NEh8).

To further strengthen our contributions and address the reviewers’ concerns, we have updated the manuscript (with all major revisions highlighted in blue) and added new analyses. Below we summarize the key changes:
- **Clarified main contribution.** We further emphasize in abstract, introduction, and conclusion that our core contribution is introducing an **explicit decision-making step** that improves the **transparency and trustworthiness** of search agents—beyond EM and efficiency.
- **Expanded empirical analyses.** We add extensive analyses and ablations on stage-2 labeling hyperparameters (jnAv, TLUY), different retrievers (TLUY), two-stage vs. end-to-end training (rGmT, jnAv), model scaling, and RL vs. SFT in Stage-2 training.  (**Section 4.3**)
- **Additional evaluations.** We provide analyses on the hyperparameters of reward-shaping baselines (jnAv; **Appendix C.2**), the average number of searches (rGmT, jnAv, TLUY; **Figure 5d**), and end-to-end latency (rGmT, TLUY; **Table 10**).
- **Reorganization for clarity.** We move AdaSearch-E2E to the appendix for cleaner presentation.  (rGmT, jnAv; **Appendix D**)
- **Improved reproducibility.** We add implementation details and training configurations.  (jnAv; **Appendix B**)
- **Case studies.** We include expanded qualitative examples for completeness. (TLUY; **Appendix F**)

**Summary:** Beyond performance and efficiency, our work tackles an overlooked but critical challenge for search agents: **transparent decision making for trustworthiness**. By introducing an **explicit decision step**, our method yields more **interpretable rationales**, improves **self-knowledge awareness**, and **reduces unnecessary search calls** while **preserving strong task performance**.

---

### Meta-Review · Area_Chair_4GmZ · 2025-12-26

**Summary:**

This paper studies reinforcement learning for training LLMs to use search tools more efficiently, aiming to avoid unnecessary search when the model’s parametric knowledge suffices. The authors prompt the model to first decide whether to rely on internal knowledge or invoke search, and then optimize this behavior with RL. The resulting approach matches the accuracy of standard search-augmented RL methods (e.g., Search-R1) while reducing search usage.

The main concern from the reviewers is the problem setting and the contribution. As analyzed below, I do not think the authors' responses fully address those concerns. Thus, I recommend rejection.

**Reviewer Concerns:**

- Reviewer rGmT has a concern about whether reducing the proportion of tasks in which search is invoked is an important setting. The author responds to this by pointing out the advantages of this setting. However, I think the concern remains outstanding, as the responses do not directly address this issue.
- Reviewer TLUY has a concern about the technical contributions, as the paper mainly re-frames existing tool-RL setups by partitioning prompts and using simple outcome-only rewards to introduce a “should I search?” The authors respond that their approach brings trustworthiness.

For both concerns, the authors list the following as arguments: no complex reward engineering, distinguishing necessary vs. unnecessary search, and explicit and interpretable decision making. However, I think both concerns are still outstanding.

**Reviewer Scores:**

Reviewer rGmT and TLUY point out two fundamental concerns. I do not think they would change their score.

---

### Decision · Program_Chairs · 2026-01-26

Reject